# A New Tensor Network: Tubal Tensor Train Network and its Applications

## Abstract

This paper introduces the Tubal Tensor Train (TTT) decomposition, a novel tensor decomposition model that effectively mitigates the curse of dimensionality inherent in the Tensor Singular Value Decomposition (T-SVD). The TTT decomposition represents an $N$-order tensor as the Tubal product (T-product) of a series of two third-order and $(N - 3)$ fourth-order core tensors, contracted. Similar to the Tensor-Train (TT) decomposition, our approach addresses the curse of dimensionality problem. In order to decompose a given tensor into the TTT format, we propose two high-performing algorithms. Numerical simulations are conducted on diverse tasks to demonstrate the efficiency and accuracy of these algorithms compared to the State-of-the-Art methods.

## 1 Introduction

Tensors are multi-dimensional arrays used to represent and manipulate complex data structures such as images, videos, and scientific datasets Sidiropoulos et al. (2017); Kolda & Bader (2009); Papalexakis et al. (2016). They have become fundamental in machine learning, computer vision, scientific computing, and engineering. Tensors are defined as a generalization of scalars, vectors, and matrices to higher dimensions. Each element in a tensor is identified by a set of indices, with each index corresponding to a specific dimension. Tensor decompositions, such as Canonical Polyadic Decomposition (CPD) Hitchcock (1927; 1928), Tucker decomposition Tucker (1963); De Lathauwer et al. (2000), Block Term decomposition De Lathauwer (2008), Tensor Train (TT) decomposition Oseledets (2011), and Tensor Chain/Ring decomposition Espig et al. (2012); Zhao et al. (2016), are used to represent tensors in more compact forms.

The Tensor Singular Value Decomposition (T-SVD) is a popular decomposition for third order tensors with applications in machine learning, signal processing, and computer vision Kilmer et al. (2013); Kilmer & Martin (2011). However, its extension to higher order tensors suffers from the curse of dimensionality Martin et al. (2013); Wang & Yang (2022). To address this issue, we propose the Tubal Tensor Train (TTT) decomposition, which extends the T-SVD to higher order tensors by breaking the curse of dimensionality Wang & Yang (2022). Unlike the Convolutional Tensor-Train model proposed in Su et al. (2020), which replaces the contraction operator with the convolutional operator LeCun et al. (1995), our approach relies on the use of the T-product Gu et al. (2018).

**Contributions and Outline of the paper**

- We introduce a novel tensor decomposition model called "Tubal Tensor Train" (TTT).
- We show that TTT model successfully mitigates the curse of dimensionality exhibited in the T-SVD model.
- We propose two efficient algorithms to compute the TTT of a higher-order tensor.
- We conduct extensive simulations to show the efficiency of our approach on diverse tasks, including tensor completion and compression of RGB images, videos, and hyperspectral images.

The paper is structured as follows. In Section 2, we introduce the necessary concepts and notations. Next, in Section 3, we discuss the characteristics and properties of the TT and T-SVD models, which are relevant for the remainder of the paper. Our proposed algorithms are described in Section 4, where we demonstrate how to overcome the curse of dimensionality that is inherent in the T-SVD model. This enhances the applicability of the T-SVD for decomposing high-order data tensors. We present the simulation results in Section 5, and finally, we provide a conclusion in Section 6.

## 2 PRELIMINARIES

This section presents the main notations, which we use throughout the paper. A tensor, a matrix, and a vector are denoted by a underlined capital letter, a bold capital letter and a bold lower case letter. The analyses and experiments done in the paper are for real-valued tensors, but they can be extended to complex tensors in a straightforward way.

We define a matrix $\mathbf{A}$ of size $I \times T$ as a *hyper-vector*, $\underline{\mathbf{a}}$, of length $I$, i.e., its elements, $\underline{\mathbf{a}}(i)$, are vectors (tubes) of length $T$. We also call this a hierarchical vector, because it has two levels of structure: the vector level and the tube level.

Similarly, a *hyper-tensor* is a tensor whose elements are tubes of the same length, which we call the tube length of the hyper-tensor. A tensor of size $I \times J \times T$ is considered an $I \times J$-dimensional *hyper-matrix* with tube length $T$. An order-$(N + 1)$ tensor is a hyper-tensor of order-$N$, where $N$ is the number of modes excluding the tube mode. For convenience, we denote the last mode of a tensor as the tube mode on which we apply the circular convolution, and use $T$ to denote the tube length of a hyper-tensor throughout the paper.

**Definition 1.** (**t-product** Kilmer & Martin (2011)) The t-product of two hyper-matrices $\underline{\mathbf{X}} \in \mathbb{R}^{I \times J}$ and $\underline{\mathbf{Y}} \in \mathbb{R}^{J \times K}$ yields a hyper-matrix $\underline{\mathbf{Z}} \in \mathbb{R}^{I \times K}$ denoted by $\underline{\mathbf{Z}} = \underline{\mathbf{X}} * \underline{\mathbf{Y}}$ whose elements are given by

$$\underline{\mathbf{Z}}(i, k) = \sum_{j=1}^{J} \underline{\mathbf{X}}(i, j) \circledast \underline{\mathbf{Y}}(j, k), \tag{1}$$

where "$\circledast$" denotes the modulo-$T$ circular convolution of tubes[1].

**Definition 2.** (**Transpose**) of a hyper-matrix $\underline{\mathbf{X}} \in \mathbb{R}^{I \times J}$ gives another hyper-matrix, denoted by $\underline{\mathbf{X}}^T \in \mathbb{R}^{J \times I}$, where $\underline{\mathbf{X}}^T(j, i) = \text{ifft}(\text{conj}(\text{fft}(\underline{\mathbf{X}}(i, j))))$, i.e., reverse the order of the 2nd to the last elements of the tube $\underline{\mathbf{X}}(i, j)$.

**Definition 3.** (**Identity hyper-matrix**) $\underline{\mathbf{I}}$ of size $I \times I$ is called identity if its off-diagonal elements are zero-tubes and its diagonal elements are unit vectors of length $T$ with the first entry equal to 1 and the rest equal to 0. This coincides with an order-3 tensor whose the first frontal slice is an identity matrix and all other frontal slices are zero. It is easy to show that $\underline{\mathbf{I}} * \underline{\mathbf{X}} = \underline{\mathbf{X}}$ and $\underline{\mathbf{X}} * \underline{\mathbf{I}} = \underline{\mathbf{X}}$ for all hyper-matrices of compatible sizes.

**Definition 4.** (**Orthogonal hyper-matrix**) A hyper-matrix $\underline{\mathbf{X}} \in \mathbb{R}^{I \times I}$ is orthogonal if $\underline{\mathbf{X}}^T * \underline{\mathbf{X}} = \underline{\mathbf{X}} * \underline{\mathbf{X}}^T = \underline{\mathbf{I}}$.

Note that there are not unique definitions for orthogonal and identity tensors and the above definitions correspond to the t-product operation.

**Definition 5.** (**tubal outer product**) of $N$ hyper-vectors, $\underline{\mathbf{a}}_n = [\underline{\mathbf{a}}_n(1); \ldots; \underline{\mathbf{a}}_n(I_n)]$ of length $I_n$ and tube length $T$, yields a rank-1 hyper-tensor, $\underline{\mathbf{Y}}$ of size $I_1 \times I_2 \times \cdots \times I_N$ with tubes of length $T$

$$\underline{\mathbf{Y}} = \underline{\mathbf{a}}_1 \circ \underline{\mathbf{a}}_2 \circ \cdots \circ \underline{\mathbf{a}}_N, \tag{2}$$

where the elements of $\underline{\mathbf{Y}}$ are given by

$$\underline{\mathbf{Y}}(i_1, i_2, \ldots, i_N) = \underline{\mathbf{a}}_1(i_1) \circledast \underline{\mathbf{a}}_2(i_2) \circledast \cdots \circledast \underline{\mathbf{a}}_N(i_N), \tag{3}$$

The tubal outer product can also be seen as a generalization of the outer product of vectors.

## 3 TENSOR SINGULAR VALUE DECOMPOSITION (T-SVD) AND TENSOR TRAIN DECOMPOSITION

In this section, we recall two tensor decomposition models, namely the *Tensor Train (TT)* decomposition and *Tensor SVD (T-SVD)*. These decompositions are crucial for the formulation of the proposed TTT model, which will be introduced in Section 4.

---

[1] For the definition of the modulo-$T$ circular convolution and a more efficient way to computate the t-product, see Appendix A.1

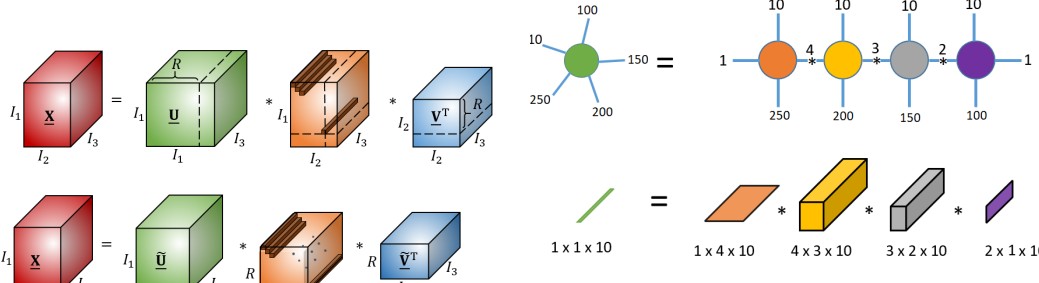

Figure 1: (Upper) The T-SVD of a third order tensor. (Bottom) The truncated T-SVD of a third order tensor.

Figure 2: (Upper) The graph visualization of the proposed tubal TT decomposition. The connection between the core tensors is the T-product. (Bottom) One tube of the original tensor is computed through a sequence of T-products between sub-tensors of the core tensors of the TTT model.

## 3.1 TENSOR TRAIN (TT) DECOMPOSITION

The Tensor-Train (TT) decomposition Oseledets (2011), also known as Matrix Product State (MPS) Hübener et al. (2010), is a powerful tool for efficient representation and manipulation of higher order tensors. It constructs a special tensor network that represents the original tensor as a train of low order tensors. Figure 10 illustrates this tensor decomposition for a 4th order tensor. Suppose $\underline{\mathbf{X}} \in \mathbb{R}^{I_1 \times \cdots \times I_N}$ is the given tensor. The TT decomposition of $\underline{\mathbf{X}}$ can be represented by the following model:

$$\underline{\mathbf{X}}(i_1, \ldots, i_N) = \sum_{r_1, \ldots, r_{N-1}=1}^{R_1, \ldots, R_{N-1}} \widehat{\underline{\mathbf{X}}}_1(i_1, r_1)\, \widehat{\underline{\mathbf{X}}}_2(r_1, i_2, r_2) \cdots \widehat{\underline{\mathbf{X}}}_N(r_{N-1}, i_N),$$

where $\widehat{\underline{\mathbf{X}}}_n \in \mathbb{R}^{R_{n-1} \times I_n \times R_n}$ for $n = 2, \ldots, N-1$ are third-order core tensors. The first and last tensors, $\widehat{\underline{\mathbf{X}}}_1$ and $\widehat{\underline{\mathbf{X}}}_N$, are second-order tensors, i.e. matrices. Moreover, the multiplet $(R_1, \ldots, R_{N-1})$ is called TT rank. The number of parameters required to represent an $N$-th order tensor $\underline{\mathbf{X}} \in \mathbb{R}^{I \times I \times \cdots \times I}$ in the TT format is $\mathcal{O}(INR^2)$. Therefore, it scales linearly with the tensor order. Notably, the best low rank TT approximation always exists and stable algorithms have been proposed to efficiently compute the TT decomposition.

## 3.2 TENSOR SVD (T-SVD)

The Tensor SVD (T-SVD) expresses a tensor as the T-product of three tensors Kilmer & Martin (2011); Kilmer et al. (2013). Given a third order tensor of size $I \times J \times T$ as a hyper-matrix $\underline{\mathbf{X}}$ of size $I \times J$, the T-SVD of $\underline{\mathbf{X}}$ is defined by $\underline{\mathbf{X}} = \underline{\mathbf{U}} * \underline{\mathbf{S}} * \underline{\mathbf{V}}^T$ where $\underline{\mathbf{U}}$ and $\underline{\mathbf{V}}$ are orthogonal hyper-matrices of size $I \times I$ and $J \times J$, $\underline{\mathbf{S}}$ is a diagonal hyper-matrix whose off-diagonal elements are zero-tubes. The tubal rank is defined as the number of nonzero fibers in $\underline{\mathbf{S}}$, see Appendix A.1 for more details.

## 4 PROPOSED TUBAL TENSOR-TRAIN DECOMPOSITION

In this section, we present the new tensor decomposition model called Tubal-TT (TTT). The main intuition behind the TTT decomposition is to express a high-dimensional data tensor as a sequence of convolution-like products of lower-dimensional core tensors. The core tensors are often of order 2, 3, or 4, and they capture the essential features and interactions of the data. The convolution-like operator used in the TTT decomposition is the tubal product, which is a generalization of the circular convolution to tensors.

The Tubal-TT decomposition represents a general hyper-tensor, $\underline{\mathbf{X}}$ of size $I_1 \times I_2 \times \cdots \times I_N$ (with tube length $T$), as a sum of rank-1 tubal tensors constructed from core hyper-tensors, $\underline{\mathbf{X}}^{(n)}$, of size

$R_{n-1} \times I_n \times R_n$, where $R_0 = R_N = 1$

$$\mathbf{X} = \sum_{r_1=1}^{R_1} \sum_{r_2=1}^{R_2} \cdots \sum_{r_{N-1}=1}^{R_{N-1}} \mathbf{X}^{(1)}(1,:,r_1) \circ \mathbf{X}^{(2)}(r_1,:,r_2) \circ \cdots \circ \mathbf{X}^{(N)}(r_{N-1},:,1), \qquad (4)$$

where the elements of $\mathbf{X}$ are given by

$$\mathbf{X}(i_1, i_2, \ldots, i_N) = \sum_{r_1=1}^{R_1} \sum_{r_2=1}^{R_2} \cdots \sum_{r_{N-1}=1}^{R_{N-1}} \mathbf{X}^{(1)}(1,i_1,r_1) * \mathbf{X}^{(2)}(r_1,i_2,r_2) * \cdots * \mathbf{X}^{(N)}(r_{N-1},i_N,1),$$

$$(5)$$

or equivalently, as the tubal product of hyper-matrices $\mathbf{X}^{(n)}(:,i_n,:)$

$$\mathbf{X}(i_1, i_2, \ldots, i_N) = \mathbf{X}^{(1)}(1,i_1,:) * \mathbf{X}^{(2)}(:,i_2,:) * \cdots * \mathbf{X}^{(N)}(:,i_N,1). \qquad (6)$$

In this sense, the TTT decomposition is interpreted as a Tubal-Matrix Product State, a tensor decomposition that operates on the tubal product. The TTT decomposition can reduce the storage and computational complexity of the data tensor as the TT model, while exploiting the convolution properties in one dimension of the data, revealing its latent structure and patterns.

To facilitate the presentation and understanding of TTT model, we start by considering a 5th-order tensor of size $100 \times 150 \times 200 \times 250 \times 10$, or a order-4 hyper tensor with tube length 10. The tensor has a TTT representation with the tubal-ranks-$(1, 4, 3, 2, 1)$ as shown in Figure 2. We use the notation $\mathbf{X} \approx \ll \mathbf{X}^{(1)}, \mathbf{X}^{(2)}, \mathbf{X}^{(3)}, \mathbf{X}^{(4)} \gg$ to represent the TTT decomposition of the tensor $\mathbf{X}$, where $\mathbf{X}^{(1)} \in \mathbb{R}^{250 \times 4 \times 10}$, $\mathbf{X}^{(2)} \in \mathbb{R}^{4 \times 200 \times 3 \times 10}$, $\mathbf{X}^{(3)} \in \mathbb{R}^{3 \times 150 \times 2 \times 10}$ and $\mathbf{X}^{(4)} \in \mathbb{R}^{2 \times 100 \times 10}$ are the core tensors.

Therefore, we have two tensors of order three at the extremities of the decomposition and two core tensors of order four. The edges connect the subsequent core tensors via the T-product. Moreover, in the bottom part of Figure 2, we show how one tube of the resulting tensor can be computed: from the first and last third-order core tensors, we extract a horizontal slice and a lateral slice, respectively, while from the two fourth-order middle tensors, we take sub-tensors (third-order tensors) and perform the T-product between them as depicted in Figure 2. Due to the similar properties of the SVD and the T-SVD, the TTT decomposition possesses the same properties as the TT decomposition in the following two senses:

- It breaks the curse of dimensionality for higher order tensors as it decomposes a tensor into core tensors of order 4 at most,
- The best TTT decomposition for a given TTT-rank is always available.

These desirable properties make the TTT decomposition of more practical interest compared to the T-SVD model. For a given TTT rank, the TTT decomposition can be computed via the TTT-SVD decomposition, which is summarized in Algorithm 1. The idea of this algorithm comes from the TT-SVD Oseledets (2011) which was proposed to decompose a tensor into the TT format. It relies on the truncated T-SVD algorithm to iteratively compute the core tensors. The key difference between the TT-SVD and the TTT-SVD is the first works on unfolded matrices, while the latter deals with reshaped form of the underlying tensors, which are of order three (see the appendix for the graphical illustration on the TTT-SVD algorithm). So, the computational complexity of this algorithm is dominated by the truncated T-SVD of third order tensors.

A fixed-precision version of the proposed algorithm can be developed by replacing the Lines 2 and 7 of Algorithm 1 by $\mathrm{T\text{-}SVD}_\delta$. Moreover, one can use the randomized algorithms developed in Zhang et al. (2018) to speed-up the computation process. An alternative way is exploiting the framework of cross or CUR approximation by sampling horizontal and lateral slices Tarzanagh & Michailidis (2018). However, our simulation results showed that similar to the TT-SVD algorithm Oseledets (2011), the TTT-SVD algorithm does not guarantee to produce a tensor with a minimum total TTT-rank or a minimum number of parameters and frequently produces models with severely unbalanced ranks. This motivated us to develop more efficient algorithms to tackle this issue, where we adopt the idea proposed Phan et al. (2020) for the TT decomposition.

Let us consider the following optimization problem

$$\min_{\mathbf{Y}} \|\mathbf{X} - \mathbf{Y}\|_F \qquad (7)$$

where $\underline{\mathbf{Y}} = \ll \underline{\mathbf{Y}}^{(1)}, \underline{\mathbf{Y}}^{(2)}, \ldots, \underline{\mathbf{Y}}^{(N)} \gg$ and has a TTT-rank $\mathbf{r} = (r_1, \ldots, r_{N-1})$. The minimization problem (7) can be solved via the Alternating Least-Squares (ALS) framework, which is a kind of block coordinate descent method. It keeps fixed all core tensors except one and it is updated by solving some least-squares problems. More advanced method is the so-called density matrix renormalization group (DMRG) technique Holtz et al. (2012); White (1993); Khoromskij & Oseledets (2010), which combines two core tensors, while keeping fixed all other core tensors. After updating the combined tensors, it is broken into two core tensors. It has been shown that the DMRG method can provide better results than the ALS approach especially we can find a tensor decomposition with a lower tensor/matrix rank easier. The TT-SVD, DMRG can compute the TT-rank based on singular values. Despite the fact that neither strategy is particularly intended for the decomposition with a particular error bound, they both aim to minimize the approximation error. These techniques work well in decompositions with known ranks or in situations where the error bound is small.

In this paper, we propose an alternative method that effectively fixes the aforementioned issues. More specifically, we consider the approach in Phan et al. (2020) as it demonstrated to deliver a TT decomposition with an optimal TT rank while tackling the problems with the TT-SVD and DMRG. We show that the TTT decomposition can be achieved by exploiting the Alternating Two-Cores Update algorithm with left-right orthogonalization (ATCU) algorithm Phan et al. (2020). The main idea is to apply the Fourier transform along the tube mode, i.e., the last dimension, of the data tensor, $\underline{\mathbf{X}}$. This gives a spectral tensor, $\hat{\underline{\mathbf{X}}} = \text{fft}(\underline{\mathbf{X}}, [], N+1)$ in the Fourier domain. Each subtensor $\underline{\mathbf{X}}(:, \ldots, :, i)$, where $i = 1, \ldots, [T/2]$ now can be decomposed into a TT-tensor using the Alternating Two-Cores Update with given approximation error bound. Then, the core tensors of the TT decomposition of different subtensors are concatenated to compute a TT decomposition of the original tensor, $\underline{\mathbf{X}}$. Finally, the core tensors are transformed back to the original space by applying the inverse FFT (IFFT). The major benefit of this method is its utilization of the ATCU algorithm, which offers improved mathematical tractability and scalability for handling large-scale data tensors. This algorithm enables the updating of one or multiple core tensors at each iteration, resulting in well-balanced TT-decompositions. Drawing inspiration from this advantageous characteristic, we have integrated the ATCU's approach for the cores update for the computation of the TTT decomposition, thus simplify the derivation of the proposed TATCU Algorithm 2.

---

**Algorithm 1:** TTT-SVD algorithm

**Input** : A hyper-tensor $\underline{\mathbf{X}}$ of size $I_1 \times I_2 \times \cdots \times I_N$ with tube length $T$ and TTT-rank $(r_1, \ldots, r_{N-1})$;

**Output:** Approximation of $\underline{\mathbf{X}}$
$$\underline{\mathbf{X}} \approx \underline{\mathbf{Y}} = \ll \underline{\mathbf{Y}}^{(1)}, \underline{\mathbf{Y}}^{(2)}, \ldots, \underline{\mathbf{Y}}^{(N)} \gg$$

1  $\underline{\mathbf{C}} = \text{reshape}\left(\underline{\mathbf{X}}, [I_1, I_2 I_3 \ldots I_N, T]\right)$
2  $[\underline{\mathbf{U}}, \underline{\mathbf{S}}, \underline{\mathbf{V}}] = \text{truncated\_TSVD}\left(\underline{\mathbf{C}}, r_1\right)$
3  $\underline{\mathbf{Y}}^{(1)} = \underline{\mathbf{U}}, \quad \underline{\mathbf{C}} = \underline{\mathbf{S}} * \underline{\mathbf{V}}^T$
4  **for** $n = 2, \ldots, N-1$ **do**
5  $\quad \underline{\mathbf{C}} = $
   $\quad \text{reshape}\left(\underline{\mathbf{C}}, [r_{n-1} I_n, I_{n+1} \ldots I_N, T]\right)$
6  $\quad [\underline{\mathbf{U}}, \underline{\mathbf{S}}, \underline{\mathbf{V}}] = \text{truncated\_TSVD}\left(\underline{\mathbf{C}}, r_n\right)$
7  $\quad \underline{\mathbf{Y}}^{(n)} = \text{reshape}\left(\underline{\mathbf{U}}, [r_{n-1}, I_n, r_n, T]\right)$
8  $\quad \underline{\mathbf{C}} = \underline{\mathbf{S}} * \underline{\mathbf{V}}^T$
9  **end**
10 $\underline{\mathbf{Y}}^{(N)} = \text{reshape}\left(\underline{\mathbf{C}}, [r_{N-1}, I_N, T]\right)$;

---

**Algorithm 2:** TATCU algorithm

**Input** : A hyper-tensor $\underline{\mathbf{X}}$ of size $I_1 \times I_2 \times \cdots \times I_N$ with tube length $T$, and a prescribed approximation error bound $\epsilon$

**Output:** Approximation
$$\underline{\mathbf{X}} \approx \underline{\mathbf{Y}} = \ll \underline{\mathbf{Y}}^{(1)}, \underline{\mathbf{Y}}^{(2)}, \ldots, \underline{\mathbf{Y}}^{(N)} \gg,$$
such that $\|\underline{\mathbf{X}} - \underline{\mathbf{Y}}\|_F \leq \varepsilon \|\underline{\mathbf{X}}\|_F$

1  FFT along tube-mode of $\underline{\mathbf{X}}$:
   $\hat{\underline{\mathbf{X}}} = \text{fft}(\underline{\mathbf{X}}, [], N+1)$
2  **for** $k = 1, 2, \ldots, T$ **do**
3  $\quad$ Decompose spectral tensor
4  $\quad \ll \underline{\mathbf{C}}^{(k_1)}, \underline{\mathbf{C}}^{(k_2)}, \cdots, \underline{\mathbf{C}}^{(k_N)} \gg =$
   $\quad \text{ATCU}(\hat{\underline{\mathbf{X}}}(:, \ldots, :, k), \epsilon)$
5  **end**
6  Glue spectral core-tensor in TT-tensors
7  **for** $j = 1, 2, \ldots, N$ **do**
8  $\quad \hat{\underline{\mathbf{Y}}}^{(j)} = [[\underline{\mathbf{C}}^{(1_j)}; \underline{\mathbf{C}}^{(2_j)}; \cdots ; \underline{\mathbf{C}}^{(I_{Nj})}]]$
9  $\quad$ Inverse FFT along tube mode
   $\quad \underline{\mathbf{Y}}^{(j)} = \text{ifft}\left(\hat{\underline{\mathbf{Y}}}^{(j)}, [], 4\right)$
10 **end**

---

Finally, we provide in Theorem 1 an upper bound on the error of the approximation computed by the TTT-SVD or the TASCU Algorithms. We assume that the error bounds of the approximations computed by the T-SVD within Algorithm 1 for the TTT-rank $(r_1, \ldots, r_{N-1})$ satisfy

$$\|\underline{\mathbf{C}} - \underline{\mathbf{U}} * \underline{\mathbf{S}} * \underline{\mathbf{V}}^T\|_F^2 \leq \delta_n^2, \tag{8}$$

where $\underline{\mathbf{U}}$, $\underline{\mathbf{S}}$, $\underline{\mathbf{V}}$ are the T-SVD factors of the hyper-matrix $\underline{\mathbf{C}}$ of tubal-rank $r_n$. Note that the hyper-matrix $\underline{\mathbf{C}}$ is an order-3 tensor at iteration $n$ of Algorithm 1.

**Theorem 1.** Let $\underline{\mathbf{X}} \in \mathbb{R}^{I_1 \times I_2 \times \cdots \times I_N}$ be a given hyper-tensor and the relation (8) holds. Then, the approximated tensor $\underline{\mathbf{Y}}$ computed by the TTT-SVD Algorithm with TTT-rank $(r_1, r_2, \ldots, r_{N-1})$ satisfies $\|\underline{\mathbf{X}} - \underline{\mathbf{Y}}\|_F^2 \leq \sum_{n=1}^{N-1} \delta_n^2$.

*Proof.* Since the SVD and T-SVD have similar properties, the proof of this theorem is similar to the one proved in Oseledets (2011) for the TT-SVD Algorithm. So, using the mathematical induction, the proof of this theorem is straightforward. □

We note that for a given tensor, the best tensor approximation with TTT-rank bounded by $r_k$ always exists since the space of all tensors of TTT-rank no higher than $r_k$ is closed. Now, using Theorem 1 and similarly to the proof of Corollary 2.4 in Oseledets (2011), it is easy to show that the following identity holds

$$\|\underline{\mathbf{X}} - \underline{\mathbf{Y}}\|_F \leq \sqrt{N-1}\|\underline{\mathbf{X}} - \underline{\mathbf{Y}}^{\text{best}}\|_F, \tag{9}$$

where $\underline{\mathbf{Y}}^{\text{best}}$ is a hyper-tensor with the best low TTT rank.

## 5 SIMULATIONS

This section presents numerical experiments to assess the proposed model compared to some baseline algorithms. All algorithms were implemented in MATLAB on a laptop computer with a 2.60 GHz Intel(R) Core(TM) i7-5600U processor and 8GB memory. The codes are available from `https://github.com/TTTmodelICLR24/TTT_MatLab_ICLR24`, and can be used to reproduce all experiments described below.

**Example 1.** (**Color images**) In this experiment, we focus on color images as real-world data tensors. The peak signal-to-noise ratio (PSNR), structural similarity index measure (SSIM) and Mean Squared Error (MSE) are used to compare the quality of the algorithms. The PSNR is defined as $\text{PSNR} = 10\log 10\left(255^2/\text{MSE}\right)$, $\text{MSE} = \|\underline{\mathbf{X}} - \underline{\mathbf{Y}}\|_F^2 / \text{num}\left(\underline{\mathbf{X}}\right)$, and "num" denotes the number of parameters of $\underline{\mathbf{X}}$, and the SSIM is defined as $\text{SSIM} = \frac{(2\mu_x\mu_y + C_1)(2\sigma_{xy} + C_2)}{(\mu_x^2 + \mu_y^2 + C_1)(\sigma_x^2 + \sigma_y^2 + C_2)}$, where $x$, $y$ are spatial patches of original and reconstructed image, $\mu_x$, $\mu_y$ are the mean intensity values of $x$ and $y$, $\sigma_x^2$, $\sigma_y^2$ are standard deviations of $x$ and $y$, $C_1$, $C_2$ are constants. The relative error is also defined as $\frac{\|\underline{\mathbf{X}} - \underline{\mathbf{Y}}\|_F}{\|\underline{\mathbf{X}}\|_F}$. We consider eight color images of size $512 \times 512 \times 3$ shown in Figure 3. We reshape the images to 10-th order tensors of size $\underbrace{4 \times \cdots \times 4}_{9} \times 3$. We use the same relative error bound of 0.15, for the TT-based model and the proposed TTT model and compute the corresponding tensors decompositions. In Table 1, the PSNR, SSIM and MSE of the reconstructed images obtained by the proposed tensor model and the TT-based approach are reported. Additionally, the reconstructed images yielded by the TT based and the proposed model are displayed in Figure 4. The proposed TTT model demonstrates significant improvements in the quality of benchmark images, as evidenced by higher PSNR and SSIM values, as well as lower MSE. It stands out by accurately preserving the background and structure of the given images. Importantly, unlike the T-SVD model which requires two tensors of the same order as the original model, our TTT model overcomes the curse of dimensionality with core tensors of order at most 4.

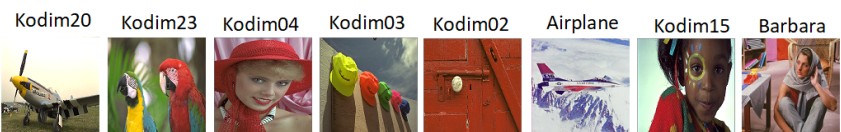

Figure 3: Benchmark images used in our experiments.

**Example 2.** (**Videos**) In this example, we consider the video datasets "Akiyo", "News", "Tempete", "Waterfall", "Foreman", and "Stephan" from `http://trace.eas.asu.edu/yuv/`. The proposed tensor model is compared with the TT decomposition and the T-SVD models in

Table 1: The experimental results obtained by the proposed algorithm and the TT-based method for images and error bound of 0.15.

| | TT-Based | | | Proposed method | | |
|---|---|---|---|---|---|---|
| Images | MSE | PSNR (dB) | SSIM | MSE | PSNR (dB) | SSIM |
| Kodim03 | 251 | 24.12 | 0.6649 | **114.09** | **27.52** | **0.7806** |
| Kodim23 | 251.40 | 24.13 | 0.7042 | **127.27** | **27.08** | **0.8096** |
| Kodim04 | 239.25 | 24.34 | 0.5940 | **93.46** | **28.42** | **0.7117** |
| Airplane | 874.30 | 18.71 | 0.6081 | **351.47** | **22.67** | **0.6921** |
| Kodim15 | 380.32 | 22.33 | 0.6178 | **159.13** | **26.11** | **0.7673** |
| Kodim20 | 436.51 | 21.73 | 0.6777 | **300.99** | **23.35** | **0.7171** |
| Barbara | 299.44 | 23.37 | 0.5746 | **121.21** | **27.30** | **0.7367** |
| Kodim02 | 141.48 | 26.62 | 0.6692 | **66.56** | **29.90** | **0.7784** |

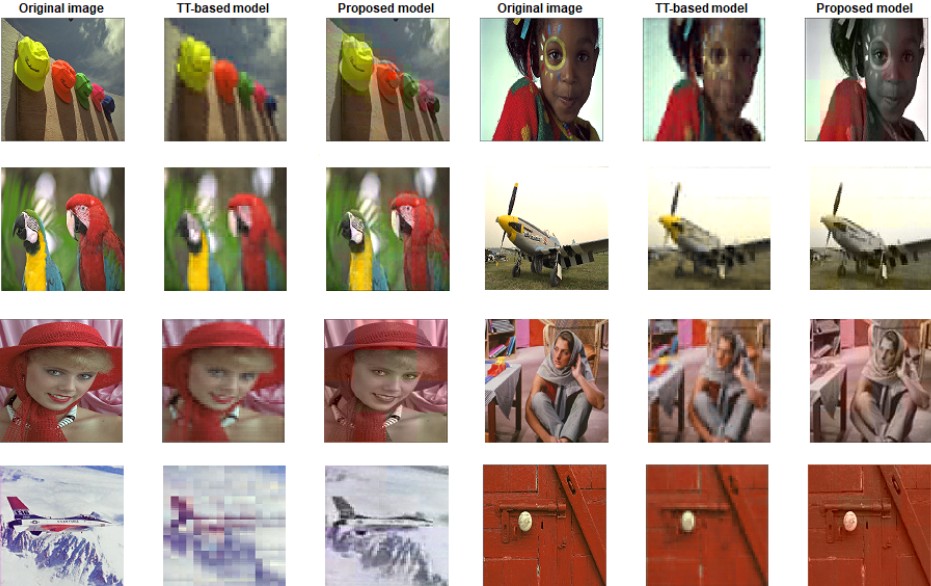

Figure 4: The reconstructed images using the proposed TTT model and the TT-based model for an upper error bound of 0.15.

terms of compression ratio and running times. The compression ratio of a tensor model is defined as $\frac{\text{number of the paramters of the tensor model}}{\text{number of the original tensor}}$, which is used in the evaluation of the algorithms. The size of all videos is $176 \times 144 \times 300$. We first reshaped the videos to 10-th order tensors of size $4 \times 4 \times 9 \times 4 \times 4 \times 11 \times 3 \times 4 \times 5 \times 5$. Our experiments are divided into two parts. In the first part we use the first three videos ("Akiyo", "News", "Tempete") and compare the efficiency of the proposed TTT model and the TT based method. To this end, we applied the proposed TTT model and the TT based decomposition with a relative approximation error bound fixed to 0.1 for the three mentioned videos. The compression ratios and running times obtained by the proposed algorithm and the TT-based are reported in Table 2. The results indicate lower running times for the proposed approach for three videos, and the compression ratio achieved by the proposed algorithm for the video "Akiyo_qci" is significantly higher than the TT-based method, while it provides a bit lower compression for the video "news_qcif". The PSNR and SSIM of all frames of the "Akiyo_qci" and "news_qcif" videos using both algorithms are shown in Figures 5. The results clearly show that the proposed TTT model can provide comparable and even better recovery results for the same relative approximation error bound in less computing time. In second experiment, we examined three videos (Waterfall", Foreman", and Stephan") using the proposed TTT model and the T-SVD model with a relative error bound of 0.1. The results are presented in Table 3. Comparing the two models, the proposed TTT model demonstrated significantly higher compression ratios but at a greater cost. These findings confirm the efficiency of the proposed tensor model for video compression task.

Table 2: The experimental results obtained by the proposed algorithm and the TT-based method.

| | TT-based model | | | Proposed method | | |
|---|---|---|---|---|---|---|
| Videos | Relative error | Running time | Compression ratio | Relative error | Running time | Compression ratio |
| News_qcif | 0.1 | 23.80 | **13.34** | 0.1 | **21.73** | 12.44 |
| Akiyo_qcif | 0.1 | 27.49 | 7.68 | 0.1 | **22.30** | **64.64** |
| Tempete_cif | 0.1 | 32.16 | 2.3283 | 0.1 | **30.33** | **2.3284** |

Table 3: The experimental results obtained by the proposed algorithm and the T-SVD model.

| | T-SVD | | | Proposed method | | |
|---|---|---|---|---|---|---|
| Videos | Relative error | Running time | Compression ratio | Relative error | Running time | Compression ratio |
| Waterfall_cif | 0.1 | 10.34 | 5.04 | 0.1 | 20.73 | **20.10** |
| Foreman_qcif | 0.1 | 9.34 | 4.42 | 0.1 | 24.56 | **11.44** |
| Stephan_qcif | 0.1 | 11.28 | 2.50 | 0.1 | 24.90 | **2.53** |

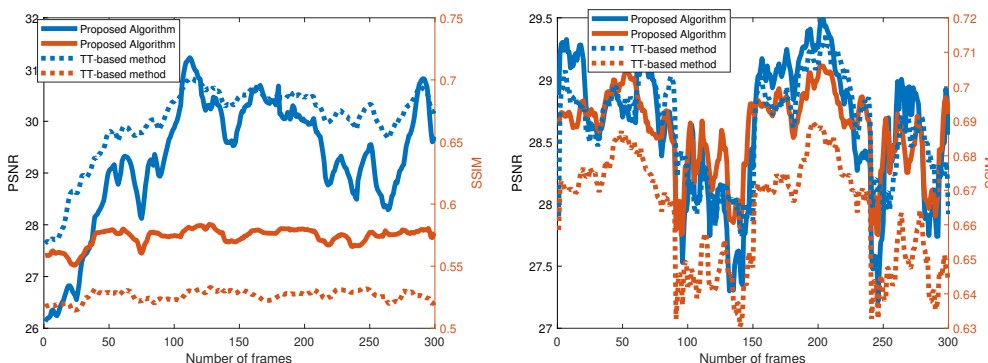

Figure 5: The PSNR and SSIM of the reconstructed frames by the proposed TTT model and TT decomposition for the relative approximation error bound of 0.1 for the "Akiyo" video (left) and the "News" video datasets.

**Example 3.** (**Application to tensor completion**) This experiment is devoted to examining the potential of the proposed model for the tensor completion task. In practice, many real-world datasets contain missing values, either due to measurement errors or incomplete data collection. Tensor completion algorithms aim to fill in these missing values by exploiting the structure of the tensor and the available observed data. The goal is to use the observed data to predict the missing values and complete the tensor. We adopt the tensor completion method developed in Ahmadi-Asl et al. (2023), where the following iterative procedure is used for the data reconstruction

$$\underline{\mathbf{X}}^{(n)} \leftarrow \mathcal{L}(\underline{\mathbf{C}}^{(n)}), \qquad (10)$$

$$\underline{\mathbf{C}}^{(n+1)} \leftarrow \underline{\mathbf{\Omega}} \circledast \underline{\mathbf{M}} + (\underline{\mathbf{1}} - \underline{\mathbf{\Omega}}) \circledast \underline{\mathbf{X}}^{(n)}, \qquad (11)$$

where $\mathcal{L}$ is an operator to compute a low-rank tensor approximation of the data tensor $\underline{\mathbf{C}}^{(n)}$, $\underline{\mathbf{1}}$ is a tensor whose all components are equal to one, $\underline{\mathbf{M}}$ is the original data tensor and the indicator set $\underline{\mathbf{\Omega}}$, stores the location of known (observed) elements. We utilize the operator $\mathcal{L}$ as an operator that computes a low tubal rank approximation computed by the T-SVD and also the low TTT rank using the proposed model. We reshaped an image to a 10-th order tensor of size $4 \times 4 \times 9 \times 4 \times 4 \times 11 \times 3 \times 4 \times 5 \times 5$ and remove 70% of the pixels. The TTT rank $[1, 2, 6, 14, 14, 14, 14, 14, 4, 1]$ was used for the TTT model and different tubal ranks were used for the T-SVD model. The best recovery results are reported for the T-SVD model. The reconstruction results for an image with 70% of pixel removed randomly are reported in Figure 6. This experiment demonstrates that for the recovery of images with missing pixels, the suggested model outperforms the T-SVD model.

**Example 4.** (**Hyperspectral Images**) This study presents a comprehensive benchmark that compares the efficiency of the proposed TTT model and associated Algorithm 2 with TT decomposition methods using hyperspectral images. The benchmark evaluates performance and decomposition quality in two settings: same accuracy ("stage 1") and same number of parameters ("stage 2"). Widely used measurements such as PSNR, RMSE, ERGAS, SAM, and UIQI are utilized. A general

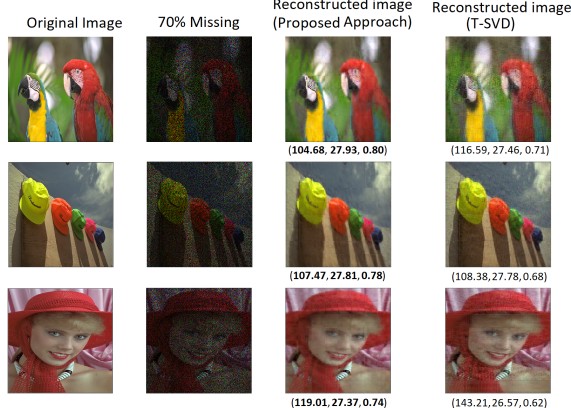

Figure 6: Illustration of the original images, images with 70% of pixels missing randomly and the reconstructed images using the proposed model and the T-SVD for Example 3. The entries represent (**MSE, PSNR, SSIM**).

Table 4: Comparison of tensor decomposition models on the ROSIS Pavia Univ. data set. The table reports the quantitative quality detailed in Section A.2.3.

| Method | Runtime (sec.) | PSNR (dB) | RMSE | ERGAS | SAM | UIQI | #Paras |
|--------|----------------|-----------|------|-------|-----|------|--------|
| Best value | 0 | $\infty$ | 0 | 0 | 0 | 1 | 0 |
| Data set - ROSIS Pavia Univ. - $N = 14$ - Relative error fixed and set to 0.08 | | | | | | | |
| TTT | 61.26 | **35.89** | **130.02** | 10.77 | 3.01 | **0.97** | **12454744** |
| TT | **42.93** | 35.89 | 130.03 | **10.71** | **2.82** | 0.97 | 13600746 |
| Data set - ROSIS Pavia Univ. - $N = 14$ - "equal" number of parameters | | | | | | | |
| TTT | 52.78 | **36.33** | **123.52** | **10.26** | 2.84 | **0.97** | 13565592 |
| TT | **42.93** | 35.89 | 130.03 | 10.71 | **2.82** | 0.97 | 13600746 |

overview of the results is reported, while detailed information about the data sets, test procedure, quality measurements, and extensive discussions of the numerical outputs can be found in Appendices A.2.1, A.2.2, A.2.3, and A.2.4, respectively. Table 4 reports the performance of each algorithm on one representative data set.

A consistent trend is observed, particularly in the specific data set being considered, but it is also noticeable across the majority of the HSI data sets. At the same level of accuracy (stage 1), the TT model consistently demonstrates faster computation time compared to the TTT model. However, the TTT model has a lower number of parameters, and both models yield similar values for the various quality measurements. As we move to stage 2, the TTT model consistently outperforms the TT model in terms of general performance. Overall, the proposed decomposition and Algorithm 2 show competitive results with the state of the art TT approach.

## 6 CONCLUSION

This paper presents a novel tensor decomposition model, called TTT, that extends the T-SVD model by effectively addressing the curse of dimensionality problem. The proposed model achieves an efficient decomposition of an $N$-order tensor into two third-order and $(N-3)$ core tensors of maximum fourth-order using the T-product. We have proposed two high-performing algorithms to decompose a given tensor into the TTT model. Extensive numerical simulations have been conducted on diverse tasks, demonstrating the efficiency of the proposed approach. In addition, our new model has the potential to harmoniously integrate with other tensor decompositions, such as CPD and TC decomposition. We are dedicating our efforts to investigating and refining these ideas as part of our ongoing research. Future works will also focus on delving into randomized variations of the proposed techniques to address the computational complexity and communication costs associated with large-scale data applications and compressing deep neural networks.

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

# A APPENDIX

## A.1 MORE PRELIMINARIES ON TENSORS

In this appendix, we provide more details about tensor concepts and definitions. Slices are subtesors obtained by fixing all but two modes of a tensor. For a third-order tensor $\underline{\mathbf{X}}$, $\underline{\mathbf{X}}(:,:,i)$, $\underline{\mathbf{X}}(:,i,:)$ and $\underline{\mathbf{X}}(i,:,:)$ are called frontal, lateral, and horizontal slices, respective. Fibers are obtained by fixing all but one mode. For a third-order tensor $\underline{\mathbf{X}}$, $\underline{\mathbf{X}}(i,j,:)$ is called a tube. $\underline{\mathbf{X}}$ can also be considered a hyper-matrix. The notation "conj" denotes the complex conjugate of a complex number or the component-wise complex conjugate of a matrix. The notation $\lceil n \rceil$ means the nearest integer number greater than or equal to $n$.

**Definition 6.** (t-product) Let $\underline{\mathbf{X}} \in \mathbb{R}^{I_1 \times I_2 \times I_3}$ and $\underline{\mathbf{Y}} \in \mathbb{R}^{I_2 \times I_4 \times I_3}$, the t-product $\underline{\mathbf{X}} * \underline{\mathbf{Y}} \in \mathbb{R}^{I_1 \times I_4 \times I_3}$ is defined as follows

$$\underline{\mathbf{C}} = \underline{\mathbf{X}} * \underline{\mathbf{Y}} = \text{fold}\left(\text{circ}\left(\underline{\mathbf{X}}\right)\text{unfold}\left(\underline{\mathbf{Y}}\right)\right), \tag{12}$$

where

$$\text{circ}\left(\underline{\mathbf{X}}\right) = \begin{bmatrix} \mathbf{X}^{(1)} & \mathbf{X}^{(I_3)} & \cdots & \mathbf{X}^{(2)} \\ \mathbf{X}^{(2)} & \mathbf{X}^{(1)} & \cdots & \mathbf{X}^{(3)} \\ \vdots & \vdots & \ddots & \vdots \\ \mathbf{X}^{(I_3)} & \mathbf{X}^{(I_3-1)} & \cdots & \mathbf{X}^{(1)} \end{bmatrix},$$

and

$$\text{unfold}(\underline{\mathbf{Y}}) = \begin{bmatrix} \mathbf{Y}^{(1)} \\ \mathbf{Y}^{(2)} \\ \vdots \\ \mathbf{Y}^{(I_3)} \end{bmatrix}, \qquad \underline{\mathbf{Y}} = \text{fold}\left(\text{unfold}\left(\underline{\mathbf{Y}}\right)\right).$$

Here, $\mathbf{X}^{(i)} = \underline{\mathbf{X}}(:,:,i)$ and $\mathbf{Y}^{(i)} = \underline{\mathbf{Y}}(:,:,i)$ for $i = 1, 2, \ldots, I_3$.

The t-product can be computed efficiently by using the Fourier transform along the third mode. More precisely, let $\widehat{\underline{\mathbf{X}}}$, $\widehat{\underline{\mathbf{Y}}}$, and $\widehat{\underline{\mathbf{Z}}}$ be the Fourier transforms of $\underline{\mathbf{X}}$, $\underline{\mathbf{Y}}$, and $\underline{\mathbf{Z}}$ along the third mode, respectively, then we have

$$\widehat{\underline{\mathbf{Z}}}(:,:,t) = \widehat{\underline{\mathbf{X}}}(:,:,t)\,\widehat{\underline{\mathbf{Y}}}(:,:,t) \tag{13}$$

for $t = 1, \ldots, T$. The inverse Fourier transform is then applied to obtain $\underline{\mathbf{Z}}$ from $\widehat{\underline{\mathbf{Z}}}$. The Fourier transform of $\underline{\mathbf{X}}$ along its third mode, which can be computed as $\widehat{\underline{\mathbf{X}}} = \text{fft}(\underline{\mathbf{X}}, [], 3)$. We refer to $\widehat{\underline{\mathbf{X}}}$ as the the transformation of the tensor $\underline{\mathbf{X}}$ in the Fourier domain. It is known that the block circulant matrix, $\text{circ}(\underline{\mathbf{X}}) \in \mathbb{R}^{I_1 I_3 \times I_2 I_3}$, can be block diagonalized, i.e.,

$$(\mathbf{F}_{I_3} \otimes \mathbf{I}_{I_1})\,\text{circ}\left(\underline{\mathbf{X}}\right)(\mathbf{F}_{I_3}^{-1} \otimes \mathbf{I}_{I_2}) = \widehat{\mathbf{X}}, \tag{14}$$

where $\mathbf{F}_{I_3} \in \mathbb{R}^{I_3 \times I_3}$ is the discrete Fourier transform matrix and $(\mathbf{F}_{I_3} \otimes \mathbf{I}_{I_1})/\sqrt{I_3}$ is a unitary matrix. Here, the block diagonal matrix $\widehat{\mathbf{X}}$ is

$$\widehat{\mathbf{X}} = \begin{bmatrix} \widehat{\underline{\mathbf{X}}}(:,:,1) & & & \\ & \widehat{\underline{\mathbf{X}}}(:,:,2) & & \\ & & \ddots & \\ & & & \widehat{\underline{\mathbf{X}}}(:,:,I_3) \end{bmatrix}, \tag{15}$$

and we have the following important properties Lu et al. (2019)

$$\widehat{\underline{\mathbf{X}}}(:,:,1) \in \mathbb{R}^{I_1 \times I_2}, \tag{16}$$

$$\text{conj}(\widehat{\underline{\mathbf{X}}}(:,:,i)) = \widehat{\underline{\mathbf{X}}}(:,:,I_3 - i + 2), \tag{17}$$

for $i = 2, \ldots, \lceil \frac{I_3+1}{2} \rceil + 1$. The t-product can be equivalently performed in the Fourier domain. Indeed, let $\underline{\mathbf{C}} = \underline{\mathbf{X}} * \underline{\mathbf{Y}}$, then from the definition of the t-product and the fact that the block circulant

matrix can be block diagonalized, we have

$$
\begin{aligned}
\operatorname{unfold}(\underline{\mathbf{C}}) &= \operatorname{circ}(\underline{\mathbf{X}}) \operatorname{unfold}(\underline{\mathbf{Y}}) \\
&= (\mathbf{F}_{I_3}^{-1} \otimes \mathbf{I}_{I_1})((\mathbf{F}_{I_3} \otimes \mathbf{I}_{I_1}) \operatorname{circ}(\underline{\mathbf{X}})(\mathbf{F}_{I_3}^{-1} \otimes \mathbf{I}_{I_2})) \\
&\quad ((\mathbf{F}_{I_3}^{-1} \otimes \mathbf{I}_{I_2}) \operatorname{unfold}(\underline{\mathbf{Y}})) \\
&= (\mathbf{F}_{I_3} \otimes \mathbf{I}_{I_1}) \widehat{\mathbf{X}} \operatorname{unfold}(\widehat{\underline{\mathbf{Y}}}),
\end{aligned}
\tag{18}
$$

where $\widehat{\underline{\mathbf{Y}}} = \operatorname{fft}(\underline{\mathbf{Y}}, [], 3)$. If we multiply both sides of equation 18 from the left-hand side with $(\mathbf{F}_{I_3} \otimes \mathbf{I}_{I_1})$, we get $\operatorname{unfold}(\widehat{\underline{\mathbf{C}}}) = \widehat{\mathbf{X}} \operatorname{unfold}(\widehat{\underline{\mathbf{Y}}})$, where $\widehat{\underline{\mathbf{C}}} = \operatorname{fft}(\underline{\mathbf{C}}, [], 3)$. This means that $\widehat{\underline{\mathbf{C}}}(:, :, i) = \widehat{\underline{\mathbf{X}}}(:, :, i) \widehat{\underline{\mathbf{Y}}}(:, :, i)$. So, it suffices to transform two given tensors into the Fourier domain and multiply their frontal slices. Then, the resulting tensor in the Fourier domain returned back to the original space via the inverse FFT. Note that due to the equations in equation 16-equation 17, half of the computations are reduced. This procedure is summarized in Algorithm 3. The tubal operations can be computed in the tensor toolbox https://github.com/canyilu/Tensor-tensor-product-toolbox.

Let $\mathbf{x}[n]$ and $\mathbf{y}[n]$ be two finite sequences of length $N$. Then, the circular convolution of two vectors $\mathbf{x}$ and $\mathbf{y}$ is denoted by $\mathbf{z}[i] = \mathbf{x}[i] \circledast \mathbf{y}[i]$ defined as follows

$$
\mathbf{z}[k] = \sum_{i=0}^{N-1} \mathbf{x}[i] \mathbf{y}[k - i],
\tag{19}
$$

It is known that the circular convolution can be efficiently computed in the Fourier domain. That is, for two vectors, $\mathbf{x}$ and $\mathbf{y}$, the circular convolution is equal to the inverse discrete Fourier transform (DFT) of the product of the vectors' DFTs.

Consider the T-SVD of a third order tensor $\underline{\mathbf{X}}$ as $\underline{\mathbf{X}} = \underline{\mathbf{U}} * \underline{\mathbf{S}} * \underline{\mathbf{V}}^T$ where $\underline{\mathbf{U}} \in \mathbb{R}^{I_1 \times I_1 \times I_3}$, $\underline{\mathbf{S}} \in \mathbb{R}^{I_1 \times I_2 \times I_3}$ and $\underline{\mathbf{V}} \in \mathbb{R}^{I_2 \times I_2 \times I_3}$. For compression purposes, one can truncate the tensor factors as $\underline{\mathbf{U}} \in \mathbb{R}^{I_1 \times R \times I_3}$, $\underline{\mathbf{S}} \in \mathbb{R}^{R \times R \times I_3}$ and $\underline{\mathbf{V}} \in \mathbb{R}^{I_2 \times R \times I_3}$, this is called the truncated T-SVD. The resulting approximation computed by the T-SVD has a significantly smaller storage and computational requirement than the original tensor, while still preserving most of the important information. Figure 1 shows an illustration of the T-SVD and truncated T-SVD. The T-SVD was initially proposed for the third-order tensors, and was later extended for higher order tensors in Martin et al. (2013). The T-SVD can be computed in the Fourier domain due to the special structure of the FFT matrices, and this shown in Algorithm 4. A fixed-precision algorithm can also be used to decompose a third order tensor in the T-SVD format. To be more specific, given an error approximation tolerance $\delta$, the algorithm finds an optimal tubal rank $R$ and the corresponding T-SVD approximation $\underline{\mathbf{Y}} \approx \underline{\mathbf{U}} * \underline{\mathbf{S}} * \underline{\mathbf{V}}^T$ such that $\|\underline{\mathbf{X}} - \underline{\mathbf{Y}}\|_F \leq \delta$, w.r.t. the Frobenius norm $\|.\|_F$. We denote this operation as $\mathrm{T\text{-}SVD}_\delta$. The T-SVD was extended to $N$-th order tensors in Martin et al. (2013), where an $N$-th order tensor $\underline{\mathbf{X}} \in \mathbb{R}^{I_1 \times I_2 \times \cdots \times I_N}$, can be represented as $\underline{\mathbf{X}} \approx \underline{\mathbf{U}} * \underline{\mathbf{S}} * \underline{\mathbf{V}}^T$ where $\underline{\mathbf{U}} \in \mathbb{R}^{I_1 \times I_1 \times I_3 \times \cdots \times I_N}$, $\underline{\mathbf{S}} \in \mathbb{R}^{I_1 \times I_2 \times I_3 \times \cdots \times I_N}$ and $\underline{\mathbf{V}} \in \mathbb{R}^{I_2 \times I_2 \times I_3 \times \cdots \times I_N}$, and the truncated T-SVD can be defined analogously. The core tensors in the T-SVD have the same order as the original data tensor, so it suffers from the curse of dimensionality.

## A.2 EXPERIMENTS ON HYPERSPECTRAL IMAGES

In this experiment, we consider hyperspectral images to evaluate the effectiveness of the proposed TTT model in comparison to the TT decomposition method. All the algorithms are implemented and tested on a laptop computer with Intel Core i7-11800H@2.30GHz CPU, and 16GB memory.

### A.2.1 DATA SETS

A hyperspectral image (HSI) is an image that contains information over a wide spectrum of light instead of just assigning primary colors (red, green, and blue) to each pixel as in RGB images. The spectral range of typical airborne sensors is 380-12700 nm and 400-1400 nm for satellite sensors. For instance, the AVIRIS airborne hyperspectral imaging sensor records spectral data over 224 continuous channels. The advantage of HSI is that they provide more information on what is imaged, some of it blind to the human eye as many wavelengths belong to the invisible light spectrum. This

---

**Algorithm 3:** Fast t-product of two tensors Kilmer & Martin (2011); Lu et al. (2019)

**Input** : Two data tensors $\underline{\mathbf{X}} \in \mathbb{R}^{I_1 \times I_2 \times I_3}$, $\underline{\mathbf{Y}} \in \mathbb{R}^{I_2 \times I_4 \times I_3}$
**Output:** t-product $\underline{\mathbf{C}} = \underline{\mathbf{X}} * \underline{\mathbf{Y}} \in \mathbb{R}^{I_1 \times I_4 \times I_3}$

1 $\widehat{\underline{\mathbf{X}}} = \text{fft}(\underline{\mathbf{X}}, [], 3)$
2 $\widehat{\underline{\mathbf{Y}}} = \text{fft}(\underline{\mathbf{Y}}, [], 3)$
3 **for** $i = 1, 2, \ldots, \lceil \frac{I_3+1}{2} \rceil$ **do**
4 $\quad \widehat{\underline{\mathbf{C}}}(:,:,i) = \widehat{\underline{\mathbf{X}}}(:,:,i) \widehat{\underline{\mathbf{Y}}}(:,:,i)$
5 **end**
6 **for** $i = \lceil \frac{I_3+1}{2} \rceil + 1, \ldots, I_3$ **do**
7 $\quad \widehat{\underline{\mathbf{C}}}(:,:,i) = \text{conj}(\widehat{\underline{\mathbf{C}}}(:,:,I_3 - i + 2));$
8 **end**
9 $\underline{\mathbf{C}} = \text{ifft}\left(\widehat{\underline{\mathbf{C}}}, [], 3\right);$

---

**Algorithm 4:** The truncated t-SVD decomposition of the tensor $\underline{\mathbf{X}}$

**Input** : The data tensor $\underline{\mathbf{X}} \in \mathbb{R}^{I_1 \times I_2 \times I_3}$ and a target tubal rank $R$
**Output:** The truncated t-SVD of the tensor $\underline{\mathbf{X}}$

1 $\widehat{\underline{\mathbf{X}}} = \text{fft}(\underline{\mathbf{X}}, [], 3)$
2 **for** $i = 1, 2, \ldots, \lceil \frac{I_3+1}{2} \rceil$ **do**
3 $\quad [\widehat{\underline{\mathbf{U}}}(:,:,i), \widehat{\underline{\mathbf{S}}}(:,:,i), \widehat{\mathbf{V}}(:,:,i)] = \text{svds}(\widehat{\mathbf{X}}(:,:,i), R)$
4 **end**
5 **for** $i = \lceil \frac{I_3+1}{2} \rceil + 1, \ldots, I_3$ **do**
6 $\quad \widehat{\underline{\mathbf{U}}}(:,:,i) = \text{conj}(\widehat{\underline{\mathbf{U}}}(:,:,I_3 - i + 2))$
7 $\quad \widehat{\underline{\mathbf{S}}}(:,:,i) = \widehat{\underline{\mathbf{S}}}(:,:,I_3 - i + 2)$
8 $\quad \widehat{\underline{\mathbf{V}}}(:,:,i) = \text{conj}(\widehat{\underline{\mathbf{V}}}(:,:,I_3 - i + 2));$
9 **end**
10 $\underline{\mathbf{U}}_R = \text{ifft}\left(\widehat{\underline{\mathbf{U}}}, [], 3\right); \underline{\mathbf{S}}_R = \text{ifft}\left(\widehat{\underline{\mathbf{S}}}, [], 3\right); \underline{\mathbf{V}}_R = \text{ifft}\left(\widehat{\mathbf{V}}, [], 3\right)$

---

additional information allows one to identify and characterize the constitutive materials present in a scenery. We consider the following real HSI:

- HYDICE Urban: The Urban data set[2] consists of $307 \times 307$ pixels and 162 spectral reflectance bands in the wavelength range 400nm to 2500nm.
- ROSIS Pavia University: The Pavia University data set[2] was acquired by the ROSIS sensor during a flight campaign over Pavia, nothern Italy, and consists of $610 \times 340$ pixels and 109 spectral reflectance bands.
- AVIRIS Moffett Field: this data sethas been acquired with over Moffett Field (CA, USA) in 1997 by the JPL spectro-imager AVIRIS [3] and consists of $512 \times 614$ pixels and 224 spectral reflectance bands in the wavelength range 400nm to 2500nm. Due to the water vapor and atmospheric effects, we remove the noisy spectral bands. After this process, there remains 159 bands.
- AVIRIS Kennedy Space Center: this data set[4] has been acquired with NASA AVIRIS instrument over the Kennedy Space Center (KSC), Florida, on March 23, 1996. AVIRIS acquires data in 224 bands of 10nm width with center wavelengths from 400nm to 2500nm. After removing water absorption and low SNR bands, 176 bands are used for the analysis. We finally extract a $250 \times 250$ subimage from this dataset for our numerical experiments.

---

[2] http://lesun.weebly.com/hyperspectral-data-set.html
[3] https://aviris.jpl.nasa.gov/data/image_cube.html
[4] http://www.ehu.eus/ccwintco/index.php?title=Hyperspectral_Remote_Sensing_Scenes

A.2.2 TEST PROCEDURE

We conduct a comprehensive benchmark of our proposed TTT model against the TT-based decomposition model for each dataset outlined in A.2.1. Note that the data sets are first folded into a $N$-order tensor before computing the tensor decompositions. The testing procedure consists of two successive and complementary stages:

1. In the first stage, we impose the same approximation error bound for both models and evaluate the performance of the approximations, as well as the number of parameters for each model. It is important to note that, for this setup, we utilize Algorithm 1 to compute the TTT decomposition.

2. In the second stage, we adapt the approximation error bound such that each model has an equal number of parameters. We then assess the quality of both decompositions under this constraint.

The overall objective is to provide a fair and comprehensive comparison of the two models.

A.2.3 PERFORMANCE EVALUATION

In order to evaluate the quality of the tensor decompositions, we consider five widely used and complementary quality measurements:

- Peak Signal-to-Noise Ratio (PSNR): PSNR assesses the spatial reconstruction quality of each band. It measures the ratio between the maximum power of a signal and the power of residual errors. A higher PSNR value indicates better spatial reconstruction quality.
- Root Mean Square Error (RMSE): RMSE is a similarity measure between the input tensor/image $\underline{\mathbf{X}}$ and the approximated tensor/image $\underline{\widehat{\mathbf{X}}}$. A smaller RMSE value indicates better fusion quality.
- Erreur Relative Globale Adimensionnelle de Synthèse (ERGAS): ERGAS provides a macroscopic statistical measure of the quality of the fused data. It calculates the amount of spectral distortion in the image Wald (2000). The best value for ERGAS is 0.
- Spectral Angle Mapper (SAM): SAM quantifies the preservation of spectral information at each pixel. It computes the angle between two vectors of the estimated and reference spectra, resulting in a spectral distance measure. The overall SAM is obtained by averaging the SAMs computed for all image pixels. A smaller absolute value of SAM indicates better quality for the approximated tensor $\underline{\widehat{\mathbf{X}}}$.
- Universal Image Quality Index (UIQI): UIQI evaluates the similarity between two single-band images. It considers correlation, luminance distortion, and contrast distortion of the estimated image with respect to the reference image Wang & Bovik (2002). The UIQI indicator ranges from -1 to 1. For multiband images, the overall UIQI is computed by averaging the UIQI values computed band by band. The best value for UIQI is 1.

For more details about these quality measurements, we refer the reader to Loncan et al. (2015) and Wei et al. (2016).

A.2.4 EXPERIMENTAL RESULTS

The performance of each algorithm is presented in Table 4 and Tables 5 to 7. It is important to note that, for the K.S.C. dataset, despite our best efforts, we were unable to achieve an equal number of parameters for both models. The TT model consistently maintained a stable number of parameters around 3e6, which was significantly lower than the number of parameters obtained for the TTT model, even at higher levels of accuracy. As a result, we modified the test procedure for this dataset by using a lower level of accuracy in stage 2. In order to give more insights on the performance comparison between models, Figures 7 to 9 display the SAM maps obtained for three first HSI data sets detailed in Section A.2 for both stages of the test procedure.

For the first three datasets, we observe a consistent trend: at the same level of accuracy (stage 1), the TT model demonstrate faster computation time compared to the TTT model. However, the TTT model exhibits a lower number of parameters, and both models yield similar values for the various quality measurements. Moving to stage 2, the TTT model consistently outperforms the TT model

overall. Analyzing the SAM maps in Figures 7 to 9, we observe that the TTT model achieves higher accuracy in areas with higher gradients. On the other hand, the TT model shows slightly better accuracy in less variable areas such as those corresponding to trees and grass (see right side of sam maps for Urban data set for instance). Notably, in Figure 9, during stage 2, the TTT model significantly outperforms the TT model in accurately reconstructing the water (northern part of the image).

In the case of the last dataset, we observe that the TTT model requires a significantly higher number of parameters to achieve both levels of accuracies. However, it is worth noting that despite this difference, the TTT model demonstrates faster computation time and yields better values for the various quality measurements.

Table 5: Comparison of tensor decomposition models on the HYDICE Urban data set. The table reports the quantitative quality detailed in Section A.2.3.

| Method | Runtime (sec.) | PSNR (dB) | RMSE | ERGAS | SAM | UIQI | #Paras |
|---|---|---|---|---|---|---|---|
| Best value | 0 | $\infty$ | 0 | 0 | 0 | 1 | 0 |
| Data set - HYDICE Urban - $N = 13$ - Relative error fixed and set to 0.08 | | | | | | | |
| TTT | 39.12 | 33.67 | **16.41** | 10.31 | 3.18 | 0.97 | **9010360** |
| TT | **22.04** | **33.80** | 16.42 | **10.08** | **3.02** | 0.98 | 10670310 |
| Data set - HYDICE Urban - $N = 13$ - "equal" number of parameters | | | | | | | |
| TTT | 23.81 | **35.92** | **12.71** | **7.95** | **2.60** | **0.98** | **10559780** |
| TT | **22.04** | 33.80 | 16.42 | 10.08 | 3.02 | 0.98 | 10670310 |

Table 6: Comparison of tensor decomposition models on the AVIRIS Moffett data set. The table reports the quantitative quality detailed in Section A.2.3.

| Method | Runtime (sec.) | PSNR (dB) | RMSE | ERGAS | SAM | UIQI | #Paras |
|---|---|---|---|---|---|---|---|
| Best value | 0 | $\infty$ | 0 | 0 | 0 | 1 | 0 |
| Data set - Moffett - $N = 10$ - Relative error fixed and set to 0.02 | | | | | | | |
| TTT | 0.68 | 39.72 | 0.01 | 4.90 | 6.22 | 1.00 | **362150** |
| TT | **0.28** | **40.61** | 0.01 | **3.10** | **5.78** | 1.00 | 418905 |
| Data set - Moffett - $N = 10$ - "equal" number of parameters | | | | | | | |
| TTT | **0.66** | **44.15** | **0.00** | **2.93** | **4.06** | 1.00 | **407975** |
| TT | **0.28** | 40.61 | 0.01 | 3.10 | 5.78 | 1.00 | 418905 |

Table 7: Comparison of tensor decomposition models on the AVIRIS Kennedy Space Center data set. The table reports the quantitative quality detailed in Section A.2.3.

| Method | Runtime (sec.) | PSNR (dB) | RMSE | ERGAS | SAM | UIQI | #Paras |
|---|---|---|---|---|---|---|---|
| Best value | 0 | $\infty$ | 0 | 0 | 0 | 1 | 0 |
| Data set - K.S.C. - $N = 14$ - Relative error fixed and set to 0.05 | | | | | | | |
| TTT | **19.39** | **39.99** | **130.13** | **86.88** | **25.16** | **0.49** | 8534545 |
| TT | 26.90 | 38.28 | 130.35 | 119.49 | 37.73 | 0.32 | **3484834** |
| Data set - K.S.C. - $N = 14$ - Relative error fixed and set to 0.02 | | | | | | | |
| TTT | **19.28** | **45.83** | **51.52** | **47.42** | **16.73** | **0.58** | 9188870 |
| TT | 25.52 | 44.14 | 52.14 | 55.26 | 25.72 | 0.42 | **3976714** |

## A.3 COMPARISON WITH THE TENSOR CHAIN MODEL

In this section, we further compare the performance of the proposed TTT model with the Tensor Chain (TC) model Espig et al. (2012); Zhao et al. (2016), which is a general form of the TT decomposition and represents a higher order data tensor as a chain of third order tensors. This is done by introducing an extra auxiliary index, which can be considered as a linear combination of the TT decomposition terms. It is known to provide more competitive results than the TT decomposition.

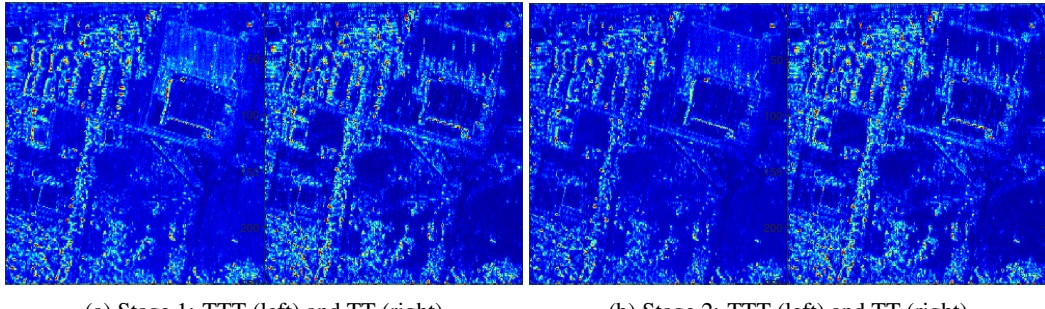

(a) Stage 1: TTT (left) and TT (right)    (b) Stage 2: TTT (left) and TT (right)

Figure 7: SAM maps for HYDICE Urban data set for both stages of the test procedure detailed in Section A.2.2, the values for sam quality measurement are limited to the interval [0;20].

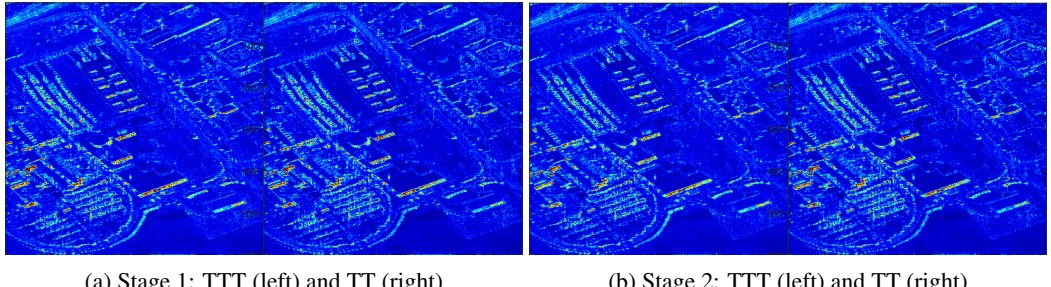

(a) Stage 1: TTT (left) and TT (right)    (b) Stage 2: TTT (left) and TT (right)

Figure 8: SAM maps for ROSIS Pavia University data set for both stages of the test procedure detailed in Section A.2.2, the values for sam quality measurement are limited to the interval [0;20].

We also note that our model can be extended to the tubal TC. To compute the TC decomposition, the codes released at the GitHub repository `https://github.com/oscarmickelin/tensor-ring-decomposition` was used. We considered the benchmark images described in Example 1 and for a given approximation error bound 0.15, we computed the TTT, TT and TC models. The PSNR achieved by the TTT, TT and TC models for the given tolerance 0.15 are reported in Table 8. We see that the TC model is relatively provides better results than the TT model but it is still not competitive with our proposed TTT model.

Table 8: The PSNR achieved by the TTT, TT and TC models for the given tolerance 0.15.

| | Images | | | | | | |
|---|---|---|---|---|---|---|---|
| Method | Airplane | Barbara | Kodim02 | Kodim03 | Kodim04 | Kodim15 | Kodim20 |
| TTT | 22.67 | 27.30 | 29.90 | 27.52 | 28.42 | 27.30 | 23.35 |
| TT | 18.71 | 23.37 | 26.62 | 24.12 | 24.34 | 22.33 | 21.73 |
| TC | 20.85 | 26.01 | 27.78 | 26.34 | 26.75 | 25.04 | 21.17 |

A.4    MORE DESCRIPTION OF THE TT DECOMPOSITION AND THE TTT-SVD ALGORITHM

In this section, we provide graphical illustrations of the TT decomposition and the TTT-SVD algorithm discussed in the paper. The graphical description of the TT model is presented in Figure 10, which shows that the TT model decomposes a tensor as a contract of a sequence of low order tensors. That is, an $N$th order tensor is decomposed as contraction of two matrices and $(N-2)$ third order tensors. The T-SVD was studied in Section 3.2, where the visualization was provided for a third order tensor in Figure 10. An alternative representation for the T-SVD as a sum of tubal tank-1 tensors is represented in Figure 11, while the Figure 12 shows the illustrations for third and fourth order tensors. This helps to better understand the T-SVD model for higher order tensors. To better describe the procedure of decomposing a tensor into the TTT model using the TTT-SVD algorithm, we have visualized Figure 13, which illustrates the details of this computation.

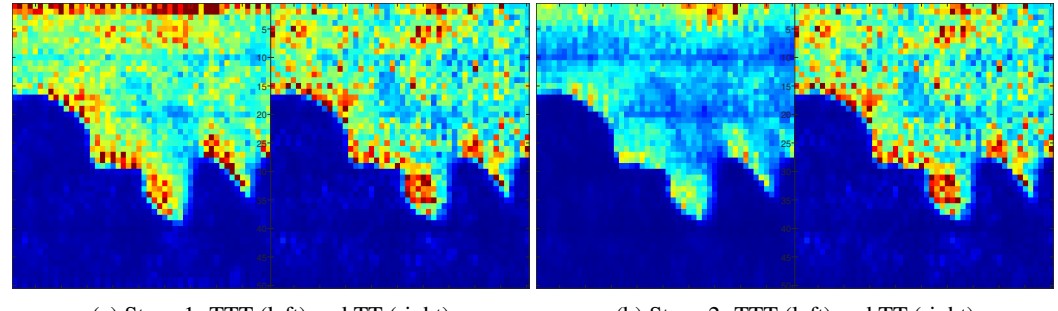

(a) Stage 1: TTT (left) and TT (right)     (b) Stage 2: TTT (left) and TT (right)

Figure 9: SAM maps for AVIRIS Moffett data set for both stages of the test procedure detailed in Section A.2.2, the values for sam quality measurement are limited to the interval [0;20].

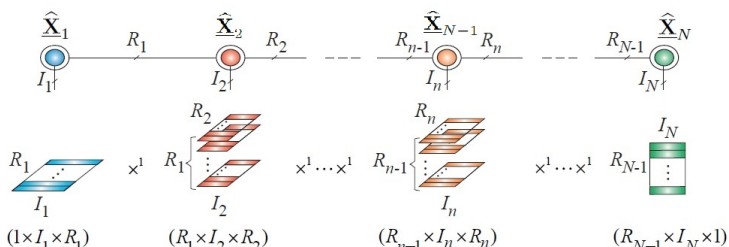

Figure 10: Structure of the TT decomposition for an $N$-th order tensor.

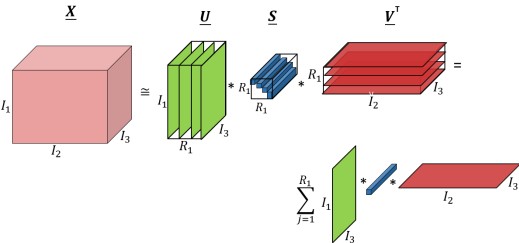

Figure 11: The truncated T-SVD of a tensor $\underline{\mathbf{X}}$ as a sum of tubal rank-1 terms.

### A.5 OTHER GENERALIZED TUBAL TENSOR MODELS

The T-SVD can be straightforwardly combined with other types of tensor decompositions such as the Tensor Ring/Tensor Chain (TR/TC) decomposition, CPD , the tensor wheel decomposition Wu et al. (2022) or an arbitrary tensor network structure. For instance, see Figure 14 a graphical illustration on the combination of the TR/TC decomposition with the T-SVD. The tubal TC/TR (TTC/TTR) model represents a higher order tensor as a collection of $N$ fourth order core tensors, see Figure 14 (Upper). A tube-wise representation of the TTC/TTR model is also demonstrated in Figure 14 (Bottom). So, a tube of the original data tensor can be approximated via the trace of the T-product of a series of tensors where trace of a third order tensor $\underline{\mathbf{X}}$ is defined as $\text{Trace}(\underline{\mathbf{X}}) = \sum_{i=1}^{I_1} \mathbf{X}(i,i,:)$. Here, the TTC/TTR rank is defined as $(r_0, r_1, \ldots, r_{N-2})$ with the condition $r_0 = r_{N-2}$, which for the case of $r_0 = r_{N-2} = 1$ is reduced to the TTT model. For the tubal HOSVD, see Wang & Yang (2022). Studying all these models is out of scope of this paper and will be the focus of our future research. Moreover, the proposed algorithms in this paper can be accelerated using the framework of randomization, which will also be investigated.

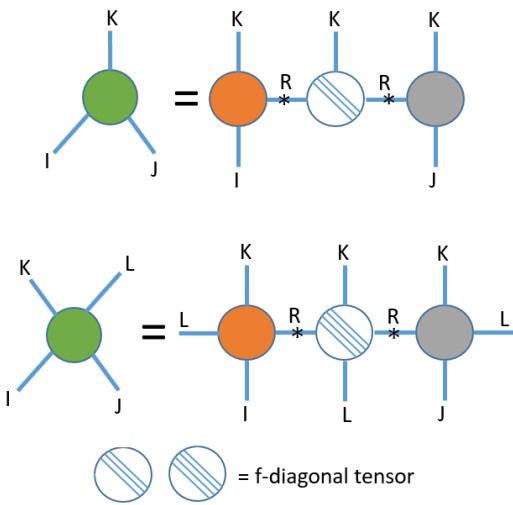

Figure 12: The truncated T-SVD of a third-order and a fourth-order tensor.

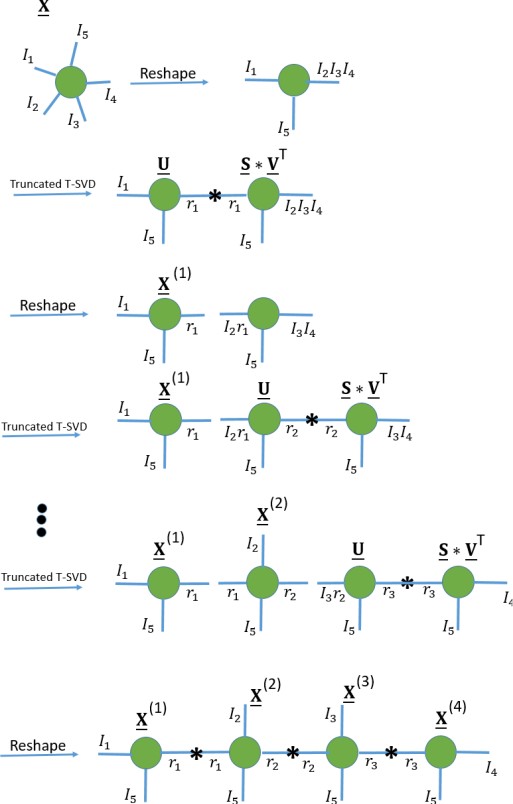

Figure 13: The procedure of the TTT-SVD for decomposing a tensor into the TTT format.

## A.6    RELATED WORKS

In this section we review recent works on the generalization of the T-SVD. The authors in Su et al. (2020) proposed to replace the contraction operator, which is used in the the TT model with convolutional operator LeCun et al. (1995); Gu et al. (2018) and they call this new model *Convolutional Tensor-Train*. They use this model to build a Long short-term memory (LSTM) network for the task

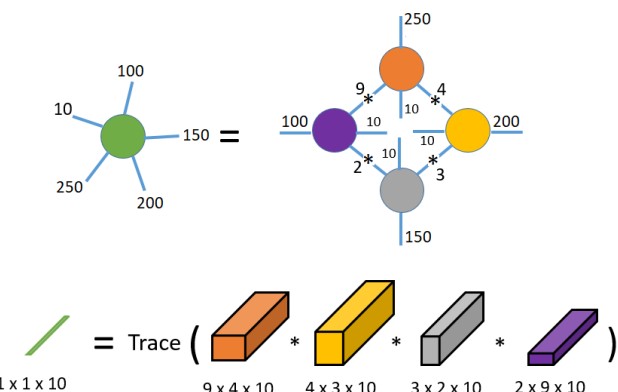

Figure 14: (Upper) The visualization of the structure of the proposed tubal TC/TR decomposition. The connection between the core tensors is the T-product. (Bottom) Computing one tubes of the original tensor is computed through a sequence of T-product of sub-tensors of the core tensors of the TTC/TTR model.

of Spatio-Temporal Learning. Our work differs from this approach as we use the t-product instead of the convolution operation. It was proposed in Wang & Yang (2022) to generalize the T-SVD to higher order tensors by introducing a new definition of the tensor transpose. Since the decomposition may be understood as a tensor-tensor product variant of the Higher Order SVD (HOSVD) De Lathauwer et al. (2000), the authors call it Hot-SVD. The existence of the Hot-SVD is proven and it is shown it is boils down to the classical T-SVD for third order tensors. The truncated and sequentially truncated versions of the Hot-SVD were also developed. The sequentially truncated version of the Hot-SVD was motivated by the Sequentially truncated HOSVD Vannieuwenhoven et al. (2012) and it was shown to be much faster. A connection between the T-SVD and HOSVD is established in Zeng & Ng (2020) by defining novel concepts such as slice rank, tensor-tensor product. These decomposition are compared by transforming the T-SVD form into the sum of outer product terms. In particular, with the use of slice rank, the sparsity of the core tensors for these two decompositions is compared. Then, inspired by the connection between the T-SVD and the HOSVD, a new decomposition called Orientation SVD (O-SVD) was proposed and the necessary theoretical and numerical analyses of the O-SVD were presented. The randomized variant of the O-SVD model was later developed in Ding et al. (2023).

