# OpenReview forum: "A New Tensor Network: Tubal Tensor Train Network and its Applications"
_ICLR.cc/2024/Conference — Submitted to ICLR 2024_

### Official Review · Reviewer_Z5ER · 2023-10-29

**Soundness:** 3 good
**Presentation:** 2 fair
**Contribution:** 1 poor
**Rating:** 5
**Confidence:** 5

**Summary:**

This work proposed a new tensor network model called TTT, where the algebra is re-defined using vectors with t-product and element-wise sum. The effectiveness of the new model is numerically verified in the task of image restoration.

**Strengths:**

The TTT model is relatively new (although I find a similar idea from https://arxiv.org/pdf/2204.10229.pdf), and introduced clearly.

**Weaknesses:**

1. Although the performance is evaluated with experiments like image compression or completion, the superior performance of TTT is not convincing. More SOTA methods should be implemented.
2. Apart from the TTT model, nothing is new compared with the original TT. It would be good if the author could highlight the uniqueness from the existing models.
3. The writing should be carefully improved. For example, Definitions 3 and 4 define the identity and orthogonal tensors in the t-product context. But it should be clarified the ambiguity from the conventionally defined identity and orthogonal tensors.

**Questions:**

No more questions for this work.

---

> ### Author Response · Authors · 2023-11-23
> **Part 1 of our answers**
>
> We thank reviewer Z5EER for reviewing our paper and raising thoughtful comments. We have prepared the answers presented below and revised the paper accordingly.
>
> 1- Although the performance is evaluated with experiments like image compression or completion, the superior performance of TTT is not convincing. More SOTA methods should be implemented.
>
> Response. Thank you for the critical comments. As we reported in the paper and the codes released on the GitHub repository, we believe that the proposed method provided better results than the T-SVD and the TT model for the compression and completion tasks. For example, for the image compression task (Example 1), the TTT model achieved an average 3 value higher PSNR for all test images compared to the TT decomposition and also the SSIM were significantly higher. For the video compression task, the TTT model achieved a much better compression ratio compared to the T-SVD and the TT model. However, to take into account the reviewer's comment, we compared our method with the with the TT since it is the main competitor. However, in the revised manuscript, we added the Tensor Chain (TC) model [2,3] in our simulation results. The TC model represents a higher order data tensor as a chain of third order tensors. This is done by introducing an extra auxiliary index, which can be considered as a linear combination of the TT decomposition terms. It is known to provide more competitive results than the TT decomposition. It it is known to provide a more compression of tensors [2,3]. We also note that our model can be extended to the tubal TC. To compute the TC decomposition, the codes released at the GitHub repository https://github.com/oscarmickelin/tensor-ring-decomposition was used. The PSNR achieved by the TTT, TT and TC models for the given tolerance 0.15 are reported in the following table.
>
>             Airplane      Barbara       Kodim02        Kodim03       Kodim04       Kodim15       Kodim20          Kodim23
>             22.67         27.30         29.90          27.52         28.42         27.30         23.35            27.30   (TTT)
>             18.71         23.37         26.62          24.12         24.34         22.33         21.73            24.13   (TT)
>             20.85         26.01         27.78          26.34         26.75         25.04         21.17            26.25   (TC)
>
> We see that the TC model is relatively provides better results than the TT model but it is still not competitive with our proposed TTT model. These results were added to an appendix in the paper. To compare with more baselines, we need more time for conducting new simulations. The PSNR achieved by the TTT, TT and TC models for a given tolerance 0.15 are reported in the following table.
>
>             Airplane      Barbara       Kodim02        Kodim03       Kodim04       Kodim15       Kodim20          Kodim23
>             22.67         27.30         29.90          27.52         28.42         27.30         23.35            27.30   (TTT)
>             18.71         23.37         26.62          24.12         24.34         22.33         21.73            24.13   (TT)
>             20.85         26.01         27.78          26.34         26.75         25.04         21.17            26.25   (TC)
>
> We see that the TC model is relatively provides better results than the TT model but it is still bot competitive with our proposed TTT model. To compare with more baselines, we need more time for conducting new simulations.
> To compare with additional baseline methods, we need more time for conducting new simulations.
>
> Moreover, we agree that the Hot-SVD model proposed in https://arxiv.org/pdf/2204.10229.pdf uses the T-Product to contract a core tensor with several unitary tubal matrices. However, this model is completely different from ours in the sense that it contract the core tensors of order at most 4 while the Hot-SVD has a core tensor of order N. The Hot-SVD still suffers from the curse of dimensionality problem since the core tensor has the same order as the original data tensor while the TTT model efficiently deals with this problem.

---

> > ### Author Response · Authors · 2023-11-23
> > **Part 2 of our answers**
> >
> > 2- Apart from the TTT model, nothing is new compared with the original TT. It would be good if the author could highlight the uniqueness from the existing models.
> >
> > Response. Thank you for the critical comments. We agree that the TTT and TT models show similarities but the TTT model significantly differs from the TT decomposition since it uses a different operation (T-Product) to contract the core tensors. Furthermore, we want to highlight that the T-SVD for higher order tensors such as fourth-order color videos, fourth-order hyper-spectral videos, fifth-order light-field images, and sixth-order bidirectional texture functions, suffers from the curse of dimensionality problem. This limits its application in these important domains. For example, in the literature, the T-SVD was used to deal with such data tensors [1] but the TTT model is more efficient and attractive on a practical point of view since it mitigates the curse of limitation of the T-SVD in an efficient way. Moreover, the tensor train (TT) with convolution contraction was investigated in [2] and competitive results were reported for learning spatio-temporal features to train an LSTM neural network. Motivated by that work and the results achieved, we proposed to replace the classical tensor contraction with the T-Product. We believe these issues and the simulations done on a variety of applications show the efficiency of the proposed tensor model and are unique contributions of the paper.
> >
> > 3- The writing should be carefully improved. For example, Definitions 3 and 4 define the identity and orthogonal tensors in the t-product context. But it should be clarified the ambiguity from the conventionally defined identity and orthogonal tensors.
> >
> > Response. Thank you for the constructive comments. We have revised the paper to make it easier to follow. We added new appendices to explain in detail the tensor concepts such as T-Product. Also, we clarified the mentioned ambiguities. Please check the revised version of Section 2  (preliminaries).
> >
> > References:
> >
> > [1] Wenjin Qin; Hailin Wang; Feng Zhang; Jianjun Wang; Xin Luo; Tingwen Huang, Low-Rank High-Order Tensor Completion With Applications in Visual Data, IEEE Transactions on Image Processing, vol. 31, pp. 2433-2448, 2022.
> >
> > [2] Jiahao Su, Wonmin Byeon, Jean Kossaifi, Furong Huang, Jan Kautz, and Anima Anandkumar. Convolutional tensor-train LSTM for spatio-temporal learning. Advances in Neural Information Processing Systems, 33:13714–13726, 2020.
> >
> > [3]  Qibin Zhao,  Guoxu Zhou, Shengli Xie, Liqing Zhang, Andrzej Cichocki, Tensor ring decomposition, arXiv:1606.05535, 2016.
> >
> > [4] Oscar Mickelin, Sertac Karaman, On algorithms for and computing with the tensor ring decomposition, Numerical Linear Algebra with Applications, vol. 27, no. 3, 2020.

---

> ### Comment · Reviewer_Z5ER · 2023-11-23
>
> Thank you to the authors for addressing my previous comments and providing additional empirical data. These results have further convinced me of TTT's superior performance in certain tasks, leading me to increase my evaluation score. However, I remain unconvinced about the aspects covered in part 2. While the authors emphasize TTT's advantages over TT and other methods, I perceive this improvement as primarily stemming from a straightforward combination of TT-product and TT. This approach doesn't seem to offer significant new insights or innovative methodologies. I would encourage a deeper exploration, perhaps considering a combination of T-product with various arbitrary tensor networks (TNs). Such an analysis could include a discussion on which specific TNs are most effective or compatible with this framework, which would add substantial depth and novelty to the paper.

---

> > ### Author Response · Authors · 2023-11-23
> > **Thanks for the new constructive comments**
> >
> > We appreciate reviewer Z5ER for new constructive comments and positive feedback on our responses. We will address the comments and reply soon.

---

### Official Review · Reviewer_agS1 · 2023-10-31

**Soundness:** 2 fair
**Presentation:** 1 poor
**Contribution:** 2 fair
**Rating:** 3
**Confidence:** 3

**Summary:**

This paper studies a new tensor decomposition model by combining tensor train (TT) decomposition and tensor SVD (T-SVD). Specifically, the authors replace tensor contractions in traditional TT with T-products defined in T-SVD. To compute the proposed TTT, the authors proposed two algorithms, which are analogous to TT-SVD and ATCU in previous literature of TT.

**Strengths:**

A new tensor decomposition is proposed with two algorithms for guaranteed low-rank approximation.

**Weaknesses:**

1. The motivation of the proposed model is not well presented. The authors stated TTT addresses the curse of dimensionality issue of T-SVD. But what is the advantage compared to TT and other tensor network structures?
2. The overall presentation is not clear enough. The preliminaries might be vague for readers not familiar with tensor decompositions. Moreover, the definition of the proposed model is not explicitly expressed. Besides, the review of related work is missing.
3. The definition of the proposed TTT is not well presented and the example in Figure 2 is not clear. I hope the authors could give a general equation of the proposed TTT, including the dimension of each factor and the meaning of each dimension.
4. For empirical evaluations, the authors only compare with TT and T-SVD, both of which are very old and classical models. It should be encouraged to choose more recent baselines.
5. The authors proposed two algorithms, TTT-SVD and TACTU, but they did not claim which one was used for experiments. Moreover, they did not mention which algorithm was used for TT.

**Questions:**

1. What is the complexity of the proposed algorithm and comparison with previous algorithms?
2. In Figure 4, it seems that TTT discards color information for Kodim 15, Barbara and Airplane images. Can the authors give some insights on this?

**Minor:**
1. The notations or preliminaries are not adequately introduced, especially for readers who are not familiar with tensors. I suppose the authors use $\ast$ to denote the T-product. However, they did not introduce this notation explicitly.
2. The authors do not clearly define the modulo-T circular convolution in Definition 1.
3. The definition of frontal slices are not introduced in Definition 5.
4. In the first Equation of Section 3.1, the middle index of $X_{N-1}$ might be $i_{N-1}$.
5. In Figure 2 bottom subfigure, is the shape of the last factor $2 \times 1 \times 10$?

---

> ### Author Response · Authors · 2023-11-23
> **Part 1 of our answers**
>
> We thank the reviewer agS1 for the detailed useful and constructive comments. We revised the paper and tried to take into account all the comments. The responses are presented below:
>
> 1- The motivation of the proposed model is not well presented. The authors stated TTT addresses the curse of dimensionality issue of T-SVD. But what is the advantage compared to TT and other tensor network structures?
>
> Response. We thank you for the critical comments. Yes, we agree that the TTT model was developed mainly motivated by the limitation of the T-SVD model in handling the curse of dimensionality. However, in contrast to the TT or TC models, the core tensors in the TTT model are contracted via the T-Product. The simulation results on image compression and completion clearly show better performance of the proposed TTT model compared to the T-SVD and the TT decomposition.  Moreover, it is worth mentioning that the TT model with convolution contraction was investigated in [2] and competitive results were reported for learning spatio-temporal features to train an LSTM neural network. Motivated by that work and the results achieved we proposed to replace the classical tensor contraction with the T-Product. We believe these issues and the simulations done on a variety of applications show the efficiency of the proposed tensor model compared to the TT model and the T-SVD are unique contributions of the paper.
>
> 2- The overall presentation is not clear enough. The preliminaries might be vague for readers not familiar with tensor decompositions. Moreover, the definition of the proposed model is not explicitly expressed. Besides, the review of related work is missing.
>
> Response. Thank you for the constructive comments. We tried to improve the presentation and make it easier for readers unfamiliar with tensor concepts. In particular, we added a new appendix A.1 to explain the T-Product more clearly and how it is related to the DFT.
>
> 3- The definition of the proposed TTT is not well presented and the example in Figure 2 is not clear. I hope the authors could give a general equation of the proposed TTT, including the dimension of each factor and the meaning of each dimension.
>
> Response. We thank you for the useful comments. We revised the paper carefully and tried to present the proposed TTT model more clearly. More information regarding the equations and the dimension of each factor were added. Please check Section 4.
>
> 4- For empirical evaluations, the authors only compare with TT and T-SVD, both of which are very old and classical models. It should be encouraged to choose more recent baselines.
>
> Response. Thank you for the critical comment. We compared our proposed model with the Tensor Chain (TC) which is known to be better than the TT model. The results still show better results of the proposed model than the TC model. Please check the Appendix A.3.
>
> 5- The authors proposed two algorithms, TTT-SVD and TACTU, but they did not claim which one was used for experiments. Moreover, they did not mention which algorithm was used for TT.
>
> Response. Thank you for your insightful feedback. We appreciate your observation. In our experiments, both TTT-SVD and TACTU algorithms were implemented. However, we found that TACTU consistently yields TTT models with significantly lower TTT-ranks compared to the TTT-SVD algorithm. Our simulation results demonstrated that, similar to the TT-SVD algorithm, TTT-SVD does not guarantee the production of a tensor with a minimum total TTT-rank or the minimum number of parameters. It often results in models with unbalanced ranks. Consequently, we exclusively reported results obtained using the TACTU algorithm, given its more favorable outcomes. Regarding the comparison with the TT model, we utilized the ACTU algorithm [put reference here], which demonstrated superior performance compared to the TT-SVD algorithm. These details, along with the reasons for our choices, have been explicitly outlined in the paper to enhance transparency and clarity in our methodology.

---

> ### Author Response · Authors · 2023-11-23
> **Part 2 of our answers**
>
> 6- What is the complexity of the proposed algorithm and comparison with previous algorithms?
>
> Response. The TTT model scales well with the order of the input tensor since it does not suffer from the curse of dimensionality in contrast to the T-SVD. For an N-th order tensor $X$ of size $I \times I \times … \times I$, and the TTT ranks $(R,R,...,R)$ the complexity of the algorithm is dominated by the DFT of the tubes which is $O(I^{N-1} \log(I))$ and the T-SVD of a third order tensors which is $O(RI^N)$. Therefore, the complexity of the algorithm is $O(I^{N-1} \log(I)+RI^N)$. Notably, the computation can be accelerated using the randomization framework or cross approximation (CUR) as previously considered  in [4] for the TT decomposition. The memory complexity of the model is $O((N-3) I^2 R^2)$ to store $(N-3)$ core tensors of size $I \times R \times I \times R$. The computational complexities of the TT decomposition and the T-SVD are $O(NRI^{N})$ and $O((N-2)I^{N-1} \log(I)+RI^N)$, respectively. The memory complexity of the TT decomposition T-SVD are also $O((N-2) R^2 I)$ and $O(I^{N-1} R)$. So, the TTT model has lower computational complexity and its memory complexity is much better than the T-SVD while it is a bit higher than the TT decomposition. Please note we have used the randomized version of all algorithms in our implementations.
>
> 7- In Figure 4, it seems that TTT discards color information for Kodim 15, Barbara and Airplane images. Can the authors give some insights on this?
>
> Response. Agreed, we acknowledge the observation. We believe that the loss of color information in specific images can be traced back to:
> 1. the reshaping process: using different reshaping strategies may affect in practice the accuracy of the approximation w.r.t. different features such as the texture, the shapes, the gradients and the colors.
> 2. As we pointed out in our responses to comments from Reviewer aedc, the differences in color information preservation probably stem from the contrasting approaches of TTT and TT when applied to color images. We think that these variations are attributed to the influence of the Discrete Fourier Transform (DFT). Indeed, in the Fourier domain, TTT tends to focus on higher-energy bands (higher magnitude coefficients), mainly within lower frequency ranges, while TT achieves a more uniform approximation of image features across various frequency bands. This emphasis on frequency components likely contributes to the divergent characteristics in feature approximation between TTT and TT.
>
> We're exploring modifications to address this discrepancy in future iterations of the TTT model, particularly focusing on adapting the methodology to better retain color information during the approximation process. One potential approach includes integrating adaptive weights tailored to different frequency bands.
>
> 8- The notations or preliminaries are not adequately introduced, especially for readers who are not familiar with tensors. I suppose the authors use * to denote the T-product. However, they did not introduce this notation explicitly.
>
> Response. Thank you for the critical comments. We agree with this comment and added more necessary information about the T-Product. More details were also added in the appendix A.1. We also tried to improve the presentation and made it easier for readers unfamiliar with tensors.
>
> 9- The authors do not clearly define the modulo-T circular convolution in Definition 1.
>
> Response. We thank you for this comment. We presented the definition of the modulo-T circular convolution in the appendix.
>
> 10- The definition of frontal slices are not introduced in Definition 5.
>
> Response. Thank you for the useful comment. We modified the definition.
>
> 11- In the first Equation of Section 3.1, the middle index of X_{N-1} might be i_{N-1}.
>
> Response. Thank you for this comment. Yes, this is true and we modified the formula.
>
> 12- In Figure 2 bottom subfigure, is the shape of the last factor 2 x 1 x 10?
>
> Response. Thank you for this comment. Yes, the size of this factor is 2 x 1 x 10 and it was an error.

---

### Official Review · Reviewer_aedc · 2023-10-31

**Soundness:** 3 good
**Presentation:** 2 fair
**Contribution:** 3 good
**Rating:** 5
**Confidence:** 5

**Summary:**

The article presents a new tensor decomposition method called Tubal Tensor Train (TTT) decomposition. The method is an extension of the Tensor Train (TT) decomposition approach, where the Tensor Singular Value Decomposition (T-SVD) and T-product methods are used to decompose N-order tensors into two third-order and (N − 3) fourth-order core tensors. The approach is claimed to  mitigate the curse of dimensionality the T-SVD method suffers. Two algorithms are discussed for the computation of the TTT decomposition.  Several numerical results are presented on different datasets and applications  to illustrate the performance of the proposed method.

**Strengths:**

Strengths:
1. A novel tensor decomposition approach is presented for higher order tensors.
2. The proposed method used the T-product and T-SVD, and a decomposition of higher order tensor with approximation guarantees.
3. The pair presents various numerical results from different applications.

**Weaknesses:**

Weakness:
1. The presentation can be improved. The paper might be hard to follow for non experts.
2. The computational cost and scalability of the proposed method, and few other details are not clear.

**Questions:**

The paper presents an interesting new tensor decomposition approach, that extends the Tensor Train (TT) method. This might be interesting in applications where there are natural multi-dimensional correlations such as videos, genetics and others.

I have the following comments about the paper:

1. The presentation is difficult to follow and certain details about the method are not clear.

i.  Firstly, the main section 4 will be difficult to follow for readers unfamiliar with tensor methods, particularly the T-product. Currently, it is not clear how the factor tensors interact with each other to form the full N-order tensor. The T-product between 4th-order tensor and how to multiply a third tensor with a fourth order tensor contracted using T-product is not obvious.

ii. t is claimed that applying T-SVD to higher order tensors suffers from the curse of dimensionality problem . But this is not clear or obvious to readers and should be explained why this is the case, and how does TTT over come this issue.

iii. how to choose the TTT-rank vector for a given N-order tensor? Are these ranks unique for given tensor? Are they related to tubal rank corresponding to T-SVD?

iv. How do we show that the space of all tensors of TTT-rank no higher than a given r_k is closed?

v. What does best low TTT rank mean and how to compute this for a given tensor?

vi. Algorithm 1 output says,  an approximation with error tolerance \epsilon is returned. But this \epsilon is not an input parameter. How do we control the error? Perhaps this is a typo.

2. The computational cost and scalability of the proposed method is not clear.

3. In the numerical experiments, we things are not clear.

i. Why is the runtime of TTT less than the TT method? Shouldn't T-SVD computation be more expensive than SVD?

ii. The advantage of TTT in the applications considered is not really clear and seems a bit forced. Reshaping the images into 10th ordertensor seems strange, and the choice of the rank values in TTT-rank vector seems arbitrary. Wouldn't considering the images (in all 4 examples) as 3rd order tensors and just using T-SVD be good enough? Comparison with this approach of just treating the images and videos in the natural 3rd order form and using T-SVD would be interesting to show that TTT actives something different/better.

iii. Why a 10-order tensor is considered and what happens to the performance if a lower order is chosen?

Minor Comment:

i. Section 3.2, middle tensor if -->  middle tensor is

ii. Recently, \star_M product, a generalization of the T-product, where FFT is replaced by a general invertible matrix M has been proposed. Perhaps, TTT can be extended using this general version of the tensor product. For image applications, it has been shown that DCT matrix as M performs better than FFT.

---

> ### Author Response · Authors · 2023-11-23
> **Part 1 of our answers**
>
> We thank the reviewer aedc for his/her effort/time to carefully read our paper and make constructive and insightful comments. We appreciate your feedbacks recognizing our contribution and novelty. We have tried to take into account all comments in the revised version of the paper. The responses to the questions are presented below:
>
> 1- The presentation can be improved. The paper might be hard to follow for non experts.
>
> Response. Thank you for the critical comment. We revised te paper carefully and tried to improve the presentation and make it easier for non expert readers. In particular, we added a new appendix to explain in detail the tensor concepts such as T-Product so that the readers can understand the topic easily, please check the appendix A. 1.
>
> 2-The computational cost and scalability of the proposed method, and few other details are not clear.
>
> Response. Thank you for the useful comment. The TTT model scales well with the order because it breaks the curse of dimensionality in contrast to the T-SVD. For an N-th order tensor $X$ of size $I \times I \times … \times I$, and the TTT ranks $(R,R,...,R)$, the complexity of the algorithm is dominated by the DFT of the tubes which is $O(I^{N-1} \log(I))$ and the T-SVD of a third order tensors which is $O(RI^N)$. SO the complexity of the algorithm is $O(I^{N-1} \log(I)+RI^N)$. This computation can be accelerated using the randomization framework or cross approximation (CUR) as was done for the TT decomposition in [4]. The memory complexity of the model is $O((N-3) I^2 R^2)$ so store $ (N-3)$ core tensors of size $I \times R \times I \times R$.
>
>
> 3- The presentation is difficult to follow and certain details about the method are not clear.
>
> Response. Thank you for the critical comment. We tried to improve the presentation and explain the proposed model more clearly.
>
> i. Firstly, the main section 4 will be difficult to follow for readers unfamiliar with tensor methods, particularly the T-product. Currently, it is not clear how the factor tensors interact with each other to form the full N-order tensor. The T-product between 4th-order tensor and how to multiply a third tensor with a fourth order tensor contracted using T-product is not obvious.
>
> Response. Thank you for the critical comment. We tried to improve the presentation and provide more information about the T-Product, please check the appendix A.1 and the revised version of the preliminary section. Regarding the second comment, please note that we don’t multiply a third order tensor with a fourth order tensor. Indeed, we multiply a third order tensor with a sub-tensor of order three a fourth order tensor.
>
> ii. it is claimed that applying T-SVD to higher order tensors suffers from the curse of dimensionality problem . But this is not clear or obvious to readers and should be explained why this is the case, and how does TTT over come this issue.
>
> Response. Thank you for your comment. The curse of dimensionality problem affects the T-SVD as it factorizes an N-th order tensor into the T-product of three tensors of order N, exacerbating computational complexity and memory requirements. In contrast, the TTT model mitigates this issue by decomposing an N-th order tensor into the T-product of core tensors, limiting their order to at most 4.
>
> iii. how to choose the TTT-rank vector for a given N-order tensor? Are these ranks unique for given tensor? Are they related to tubal rank corresponding to T-SVD?
>
> Response. Thank you for your insightful comment. Much like the TT-rank or the Tucker rank, determining the optimal TTT-rank poses a challenge. Similar to TT-ranks, TTT-ranks are not unique. It's worth noting that the TTT-rank is associated with the tubal rank, and the elements of TTT-ranks represent the tubal rank of the reshaped form of the underlying tensors, as illustrated in Figure 13 for further clarification.
>
>
> iv. How do we show that the space of all tensors of TTT-rank no higher than a given r_k is closed?
>
> Response. It is known that the collection of tensors of TT-rank no higher than r_k forms a closed subset [1]. The tensors in the TTT format have a similar structure as the TT decomposition where instead of the SVD, the T-SVD is used. On the other hand, the T-SVD and SVD have similar optimality properties. So from these two points, it can be easily proved that the space of tensors of TTT-rank no higher than a given r_k is closed.
>
> v. What does best low TTT rank mean and how to compute this for a given tensor?
>
> Response. The best low TTT rank is defined as the following optimization problem
>
> $\min   ||X-Y||_F$    subject to  TTT-rank(Y)=R
>
> where $Y$ is in the TTT format. This minimization problem can be solved via Alternating Least Squares (ALS) or the density matrix renormalization group (DMRG) technique.

---

> > ### Author Response · Authors · 2023-11-23
> > **Part 2 of our answers**
> >
> > vi. Algorithm 1 output says, an approximation with error tolerance \epsilon is returned. But this \epsilon is not an input parameter. How do we control the error? Perhaps this is a typo
> >
> > Response. We thank you for this comment. Yes, we agree, this is a typo. The $\epsilon$ will be removed.
> >
> > 4- The computational cost and scalability of the proposed method is not clear.
> >
> > Response. In our simulation results, the runtime of the TTT model was observed to be less than that of the TT model. This discrepancy can be attributed to the fact that each iteration of the TTT model necessitated a comparatively lower TTT rank than the TT model. The reduced complexity in determining the TTT rank contributes to the overall shorter runtime in our specific simulations.
> >
> >
> > ii. The advantage of TTT in the applications considered is not really clear and seems a bit forced. Reshaping the images into 10th order tensor seems strange, and the choice of the rank values in TTT-rank vector seems arbitrary. Wouldn't considering the images (in all 4 examples) as 3rd order tensors and just using T-SVD be good enough? Comparison with this approach of just treating the images and videos in the natural 3rd order form and using T-SVD would be interesting to show that TTT actives something different/better.
> >
> > Response. Thank you for your insightful comment. Preserving the multidimensional structures of images and videos is paramount. The reshaping into 10th order tensors aims not to underscore the value of reshaping alone but to simulate real-life scenarios involving high-order input tensors. This allows us to assess tensor networks' performance in handling complex data structures. We acknowledge the suggestion to compare TTT with T-SVD on the natural 3rd order form. This comparison will be incorporated into our analysis to provide a more comprehensive understanding of the differences between TTT and T-SVD in these applications.
> > Furthermore, the usefulness of reshaping in achieving higher compression ratios has been established in prior research [2,3]. Notably, our primary focus is not on the reshaping technique, leveraging well-known efficient methods to achieve this.
> > We aim to delve into the contrasting approximation approaches of TTT and TT when applied to color images, attributing these differences to the influence of the Discrete Fourier Transform (DFT). In the Fourier domain, the TTT method notably accentuates and approximates higher-energy bands, predominantly situated within lower frequency ranges. Conversely, TT achieves a more uniform approximation of image features across various frequency bands. This distinct emphasis on frequency components likely underlies the divergent characteristics in feature approximation between TTT and TT. Our future works will involve proposing modified versions of the TTT model to address this discrepancy. One potential approach involves integrating adaptive weights tailored to different frequency bands.
> >
> > iii. Why a 10-order tensor is considered and what happens to the performance if a lower order is chosen?
> >
> > Response. We have considered a tensor of high order to better show the performance of the TTT model, which is well suited for tensors of high order. We should emphasize that for tensors of lower order the performance of the TTT model compared to the T-SVD is reduced.
> >
> > Minor Comment:
> > i. Section 3.2, middle tensor if --> middle tensor is
> >
> > Response. Thank you for finding this typo. We corrected it.
> >
> > ii. Recently, \star_M product, a generalization of the T-product, where FFT is replaced by a general invertible matrix M has been proposed. Perhaps, TTT can be extended using this general version of the tensor product. For image applications, it has been shown that DCT matrix as M performs better than FFT.
> >
> > Response. Thank you for the useful suggestions. We will incorporate the generalized T-product and DCT operations in our future works while comments on these issues will be added in the paper.
> >
> > References:
> >
> > [1] André Uschmajew, Bart Vandereycken, Geometric Methods on Low-Rank Matrix and Tensor Manifolds, Handbook of Variational Methods for Nonlinear Geometric Data pp 261–313, 2020.
> >
> > [2] Anh-Huy Phan, Andrzej Cichocki, Andre Uschmajew, Petr Tichavsky, George Lutaand  Danilo Mandic, Tensor networks for latent variable analysis: Novel algorithms for tensor train approximation, IEEE transactions on neural networks and learning systems, vol. 31, no. 11, pp. 4622--4636.
> >
> > [3] Ivan Oseledets, Tensor-train decomposition, SIAM Journal on Scientific Computing, vol.33, No.5, pp. 2295--2317, 2011.
> >
> > [4] Ivan Oseledets, Eugene Tyrtyshnikov, TT-cross approximation for multidimensional arrays, Linear Algebra and its Applications, vol. 432, no.1 , pp. 70--88, 2010.

---

> ### Comment · Reviewer_aedc · 2023-12-01
>
> I thank the authors for their thorough responses to all reviewers' comments. I agree with the other reviewers that the certain aspects (novelty, cost, comparisons) of the paper can be further improved. I am keeping my score.

---

### Official Review · Reviewer_qk5Z · 2023-11-05

**Soundness:** 2 fair
**Presentation:** 3 good
**Contribution:** 2 fair
**Rating:** 6
**Confidence:** 3

**Summary:**

This paper introduces a novel tensor decomposition model called the Tubal Tensor Train (TTT) and shows that it successfully mitigates the curse of dimensionality exhibited in the Tensor Singular Value Decomposition (T-SVD) model. The paper proposes two efficient algorithms to compute the TTT of an input higher-order tensor and conducts extensive simulations to show the efficiency of the approach on diverse tasks.

**Strengths:**

1. This paper introduces a new tensor decomposition model called the TTT, which mitigates the curse of dimensionality exhibited in the T-SVD model.

2. This paper proposes two efficient algorithms to compute the TTT of an input higher-order tensor, one is a fixed-rank version, the other is a fixed-precision version.

3. The proposed TTT model is shown to outperform TT-based model in terms of efficiency and accuracy on diverse tasks.

**Weaknesses:**

1. It is better to show an intuition behind TTT decomposition for better understanding.

2. In theory, why can TTT decomposition surpass TT decomposition? The theoretical analysis is insufficient.

3. It is better to report the error bar for the experimental results.

4. It is better to provide more SOTA baseline methods, rather than just TT decomposition.

5. The notation $\ast$ in Section 2 Definition 3 does not specify.

6. $I_N$ lost $I$ in the last few lines of Section 3.2.

**Questions:**

1. Why the compression ratio of the proposed method is lower than that of TT-based model in “News qcif” dataset?

2. You claim that “The key difference between the TT-SVD and the TTT-SVD is the first works on unfolded matrices, while the latter deals with reshaped form of the underlying tensors, which are of order three”. Thus, can a decomposition deals with reshaped form of the underlying tensors with order greater than three achieve even better performance?

---

> ### Author Response · Authors · 2023-11-23
> **Part 1 of our answers**
>
> We appreciate the reviewer qk5z for reading the paper very carefully and providing many valuable comments and suggestions. We have revised the paper according to the reviewer’s comments and here are our main responses:
>
>  1- It is better to show an intuition behind TTT decomposition for better understanding.
>
> Response. Thank you for the insightful comment.
> The main intuition behind the TTT decomposition is to express a high-dimensional data tensor as a sequence of convolution-like products of lower-dimensional core tensors. The core tensors are often of order 2, 3, or 4, and they capture the essential features and interactions of the data. The convolution-like operator used in the TTT decomposition is the tubal product, which is a generalization of the circular convolution to tensors.
> To illustrate the TTT decomposition, we first introduce some new concepts and notations. We define a matrix of size $I \times T$ as a hyper-vector of length $I$, i.e., its elements are vectors (tubes) of length $T$. We call this a hierarchical vector, because it has two levels of structure: the vector level and the tube level. We also define the tubal-outer product of hyper-vectors, $a_n$ of length $I_n$ and tube length $T$, which yields a hyper-tensor, $Y$ of size $I_1 \times I_2 \times \cdots \times I_N$
> $$Y = a_1 \ast a_2 \ast \cdots \ast a_N$$
> where the elements of $Y$ are tubes of length $T$ given by
> $$Y(i_1,i_2, \ldots, i_N) = a_1(i_1) \circledast a_2(i_2) \circledast \cdots \circledast a_N(i_N)$$
> where $\circledast$ denotes the modulo-$T$ circular convolution of tubes.
> The hyper-tensor $Y = a_1 \ast a_2 \ast \cdots \ast a_N$ represents a rank-1 tubal tensor, which is the simplest form of a tubal tensor. The TTT decomposition represents a general tubal tensor, $Y$, as a sum of rank-1 tubal tensors constructed from core hyper-tensors, $A_n$, of size $R_{n} \times I_n \times R_{n+1}$, where $R_1 = R_{N+1} = 1$
> $$Y(i_1,i_2, \ldots, i_N) = \sum_{r_1 = 1}^{R_1} \sum_{r_2 = 1}^{R_2} \cdots \sum_{r_{N+1} = 1}^{R_{N+1}} A_1(r_1,i_1,r_2) \ast A_2(r_2,i_2,r_3) \ast \cdots \ast A_N(r_{N},i_N,r_{N+1})$$
> or equivalently, as the tubal product of hyper-matrices $A_n(:,i_n,:)$:
> $$Y(i_1,i_2, \ldots, i_N) =  A_1(1,i_1,:) \ast A_2(:,i_2,:) \ast \cdots \ast A_N(:,i_N,1)$$
> In this sense, the TTT decomposition is interpreted as a Tubal-Matrix Product State, a tensor decomposition that operates on the tubal product. The TTT decomposition can reduce the storage and computational complexity of the data tensor as the TT model, while exploiting the convolution properties in one dimension of the data, revealing its latent structure and patterns.
>
> 2- In theory, why can TTT decomposition surpass TT decomposition? The theoretical analysis is insufficient.
>
> Response: The TTT decomposition can surpass the TT decomposition when the data tensor has a convolutive structure in one of its modes, which means that the latent features are mixed by the circular convolution instead of the linear combination. The TTT decomposition preserves the mode where the circular convolution operates, and decomposes the other modes into lower-dimensional core tensors. This way, the TTT decomposition can capture the convolutive feature patterns of the data tensor more effectively, and provide better reconstruction and compression.
> The TT decomposition, on the other hand, decomposes all the modes of the data tensor into lower-dimensional core tensors, regardless of the structure of the data. This can result in a loss of information and accuracy, especially when the data tensor has a high order or a large rank. The TT decomposition assumes that the latent features are linearly combined in all the modes, which may not be true for some types of data.
> Therefore, the TTT decomposition can surpass the TT decomposition when the data tensor admits the TTT model, which is more suitable for data with a convolutive structure in one of its modes.
>
> 3- It is better to report the error bar for the experimental results.
>
> Response. Thank you for the suggestion. The Mean Squared Errors were reported for the image compression task (Example). We will report the errors in our experimental results. However, the relative errors are reported below fo images.As can be seen, in all cases the proposed TTT model achieves a lower relative error.
>
>                Airplane     Barbara      Kodim02     Kodim03      Kodim04     Kodim15      Kodim20      Kodim23
>               1.082e-01    1.496e-01    1.500e-01    1.466e-01    1.460e-01   1.485e-01    1.012e-01    1.497e-01 (TTT)
>               1.499e-01    1.500e-01    1.500e-01    1.510e-01    1.510e-01   1.499e-01    1.497e-01    1.500e-01 (TT)

---

> ### Author Response · Authors · 2023-11-23
> **Part 2 of our answers**
>
> 4- It is better to provide more SOTA baseline methods, rather than just TT decomposition.
>
> Response. We thank you for the constructive comment. We compare the proposed model with the TT since it is the main competitor. However, in the revised manuscript, we added the Tensor Chain (TC) model [2,3] in our simulation results. The TC model represents a higher order data tensor as a chain of third order tensors. This is done by introducing an extra auxiliary index, which can be considered as a linear combination of the TT decomposition terms. It is known to provide more competitive results than the TT decomposition. It it is known to provide a more compression of tensors [2,3]. We also note that our model can be extended to the tubal TC. To compute the TC decomposition, the codes released at the GitHub repository https://github.com/oscarmickelin/tensor-ring-decomposition was used. The PSNR achieved by the TTT, TT and TC models for the given tolerance 0.15 are reported in the following table.
>
>             Airplane      Barbara      Kodim02      Kodim03      Kodim04      Kodim15      Kodim20      Kodim23
>             22.67          27.30        29.90       27.52        28.42        27.30        23.35        27.30    (TTT)
>             18.71          23.37        26.62       24.12        24.34        22.33        21.73        24.13    (TT)
>             20.85          26.01        27.78       26.34        26.75        25.04        21.17        26.25    (TC)
>
>  We see that the TC model is relatively provides better results than the TT model but it is still bot competitive with our proposed TTT model. These results were added to an appendix in the paper. To compare with more baselines, we need more time for conducting new simulations.
>
> 5- The notation * in Section 2 Definition 3 does not specify.
>
> Response. We thank you for highlighting this issue. We modified this part and solved this problem.
>
> 6-  $I_N$ lost $I$ in the last few lines of Section 3.2.
>
> Response. Thank you for finding this inconsistency. We solved this problem.
>
> 7-IN lost I in the last few lines of Section 3.2.
>
> Response. Thank you very much for carefully reading the paper. We corrected this issue.
>
> 8- Why the compression ratio of the proposed method is lower than that of TT-based model in “News qcif” dataset?
>
> Response. While the proposed model may not consistently outperform the TT model, as demonstrated in this specific instance where the TT model yielded superior results, it's central to highlight that the compression ratio of different methods can vary for video compression. Several factors, including codec variations, video content type, and quality settings, contribute to these variations. The TT model and the TTT decomposition employ distinct contraction operators between core tensors, leading to different compression approaches. It raises an intriguing question: what factors contribute to the TTT model's lower compression ratio for the New QCIF model? Investigating this aspect could be a valuable avenue for future research, albeit one that demands more time and effort.
>
> 9- You claim that “The key difference between the TT-SVD and the TTT-SVD is the first works on unfolded matrices, while the latter deals with reshaped form of the underlying tensors, which are of order three”. Thus, can a decomposition deals with reshaped form of the underlying tensors with order greater than three achieve even better performance?
>
> Response. Yes, we agree that the TT-SVD and TTT-SVD are closely related to each other where they are working on different reshaped forms of the underlying tensors. However, it is important to recall that the contract operation between core tensors in the T-SVD and the TT model are different. The simulation results show that it is possible to achieve better results than the TT model.
>
> References:
>
> [1] Maolin Che, Yimin Wei, Hong Yan, The computation of low multilinear rank approximations of tensors via power scheme and random projection, SIAM Journal on Matrix Analysis and Applications, vol. 41, pp. 605-636, 2020.
>
> [2]  Qibin Zhao,  Guoxu Zhou, Shengli Xie, Liqing Zhang, Andrzej Cichocki, Tensor ring decomposition, arXiv:1606.05535, 2016.
>
> [3] Oscar Mickelin, Sertac Karaman, On algorithms for and computing with the tensor ring decomposition, Numerical Linear Algebra with Applications, vol. 27, no. 3, 2020.

---

> > ### Comment · Reviewer_qk5Z · 2023-11-23
> >
> > Thank you for your detailed response. I appreciate your efforts in addressing my concerns. I have read your response and comments of other reviewers. I will keep my score.

---

### Author Response · Authors · 2023-11-23
**Thanks reviewers, for your constructive comments. (The PDF version will be updated soon)**

Dear reviewers

We thank you for your time and effort in reading our paper carefully and proving many insightful and useful comments, which allowed us to improve the paper significantly. We have tried to take all comments into account. Our responses are presented below, and the updated or revised version of our paper will be uploaded soon. We apologize for the delay in uploading the revised version of our paper.

---

### Meta-Review · Area_Chair_2GnL · 2023-12-05

**Metareview:**

This paper introduces a new tensor decomposition technique called Tubal Tensor Train (TTT) decomposition. The paper developed efficient algorithm for TTT decomposition and demonstrated that TTT decomposition outperform tensor-train (TT) decomposition. Most reviewers find the idea of TTT decomposition interesting, however there are some concerns about the theoretical justifications, clarity of presentation and cost of the algorithms. The authors have addressed several concerns during the response, but several reviewers believe that the paper would benefit from major further improvements.

**Justification For Why Not Higher Score:**

It looks like a promising paper but the reviewers are not convinced that the updated version addresses all concerns.

**Justification For Why Not Lower Score:**

N/A

---

### Decision · Program_Chairs · 2024-01-16

Reject